# BayOTIDE: Bayesian Online Multivariate Time series Imputation with functional decomposition

## Abstract

In real-world scenarios like traffic and energy, massive time-series data with missing values and noises are widely observed, even sampled irregularly. While many imputation methods have been proposed, most of them work with a local horizon, which means models are trained by splitting the long sequence into batches of fit-sized patches. This local horizon can make models ignore global trends or periodic patterns. More importantly, almost all methods assume the observations are sampled at regular time stamps, and fail to handle complex irregular sampled time series arising from different applications. Thirdly, most existing methods are learned in an offline manner. Thus, it is not suitable for many applications with fast-arriving streaming data. To overcome these limitations, we propose *BayOTIDE* : Bayesian Online Multivariate Time series Imputation with functional decomposition. We treat the multivariate time series as the weighted combination of groups of low-rank temporal factors with different patterns. We apply a group of Gaussian Processes (GPs) with different kernels as functional priors to fit the factors. For computational efficiency, we further convert the GPs into a state-space prior by constructing an equivalent stochastic differential equation (SDE), and developing a scalable algorithm for online inference. The proposed method can not only handle imputation over arbitrary time stamps, but also offer uncertainty quantification and interpretability for the downstream application. We evaluate our method on both synthetic and real-world datasets.

## 1 Introduction

Multivariate time series data are ubiquitous and generated quickly in many real-world applications Jin et al. (2023), such as traffic Li et al. (2015) and energy Zhu et al. (2023). However, the collected data are often incomplete and noisy due to sensor failures, communication errors, or other reasons. The missing values in the time series data can lead to inaccurate downstream analysis. Therefore, it is essential to impute the missing values in the time series data in an efficient way.

Early methods for multivariate time series imputation Acuna & Rodriguez (2004); Van Buuren & Groothuis-Oudshoorn (2011); Durbin & Koopman (2012) are mainly based on statistical models. DNN-based imputation methods get boosted attention Fang & Wang (2020) in recent years, for their ability to model complex non-linear patterns. Another remarkable direction is to apply diffusion models Song et al. (2020); Ho et al. (2020) to handle probabilistic imputation, where filling the missing value can be modeled as a denoising process. Most recent work TIDER LIU et al. (2022) proposed another imputation direction to apply the matrix factorization and decompose the series into disentangled representations.

Despite the success of the proposed methods, they are still limited in several aspects. First, most DNN-based and diffusion-based methods are trained by splitting the long sequence into small patches. This local horizon can fail to capture the crucial global patterns Alcaraz & Strodthoff (2022); Woo et al. (2022), such as trends and periodicity, leading to less interpretability. Second, many methods assume the observations are sampled at regular time stamps, and always under-utilize or ignore the real timestamps. Thus, those models can only impute on fixed-step and discretized time points, instead of the arbitrary time stamps at the whole continuous field. Lastly, in real-world applications

| Properties / Methods | **BayOTIDE** | TIDER | Statistic-based | DNN-based | Diffusion-based |
|---|---|---|---|---|---|
| Uncertainty-aware | ✓ | ✗ | ✗ | ✗ (partial) | ✓ |
| Interpretability | ✓ | ✓ | ✓ | ✗ | ✗ |
| Continuous modeling | ✓ | ✗ | ✗ | ✗ | ✗ (partial) |
| Inference manner | **online** | offline | offline | offline | offline |

Table 1: Comparison of *BayOTIDE* and main-stream multivariate time series imputation methods. ✗ means the property can only be achieved by partial models in the family, or is not clear in the original paper. For example, only deep models with probabilistic modules can offer uncertainty quantification, such as GP-VAE Fortuin et al. (2020), but most deep models are deterministic. The diffusion-based CSDI Tashiro et al. (2021) and CSBI Chen et al. (2023) take timestamps as input, but the model is trained with discretized time embedding.

such as electricity load monitoring, massive time series data are generated quickly and collected in a streaming way Liu et al. (2023). It is extremely expensive or even impossible to retrain the model from scratch when a new data point arrives. Thus, to align with streaming data, the imputation model should work and update in an efficient online manner. However, to the best of our knowledge, all prior imputation methods are designed and optimized in an offline manner, i.e., go through all collected data several epochs to train, which is not suitable for streaming data scenarios.

To handle the above limitations, we propose *BayOTIDE* : Bayesian Online multivariate Time series Imputation with functional DEcomposition. *BayOTIDE* treats the observed values of multivariate time series as the noisy samples from a temporal function, and the goal is to estimate the whole function with uncertainty in the continuous time field. To this end, we decompose the function into a weighted combination of groups of temporal factors. Each factor is a function aimed to capture the interpretable pattern globally. We apply a group of Gaussian Processes (GPs) with smooth and periodic kernels as functional priors to fit the factors. Emploring the SDE representation of GPs and advanced moment-matching techniques, we develop an online algorithm to infer the running posterior of weights and factors efficiently. As it is a Bayesian model, *BayOTIDE* can offer uncertainty quantification and robustness against noise. The learned functional factors can provide not only interpretability but also imputation over arbitrary time stamps. We list the comparison of *BayOTIDE* and other main-stream imputation methods in Table 1. In summary, we highlight our contributions as follows:

- We propose *BayOTIDE* , a novel Bayesian method for multivariate time series imputation. *BayOTIDE* can explicitly learn the function factors representing various global patterns, which offer interpretability and uncertainty quantification. As *BayOTIDE* is a continuous model, it can utilize the irregularly sampled timestamps and impute over arbitrary time stamps naturally.

- To the best of our knowledge, *BayOTIDE* is the first probabilistic imputation method of multivariate time series that works in online mode for streaming data. Furthermore, we develop a scalable online algorithm that bridges GPs with SDE for efficient inference.

- We extensively evaluate our method on both synthetic and real-world datasets, and the results show that *BayOTIDE* outperforms the state-of-the-art methods in terms of both accuracy and uncertainty quantification.

## 2 RELATED WORK

**Disentangled representations of time series.** The most classical framework of decomposing time series into disentangled representations is the seasonal-trend decomposition(STL) (Cleveland et al., 1990) along with its following work (Wen et al., 2019; Abdollahi, 2020; Bandara et al., 2021), which are non-parametric method to decompose the univariate series into seasonal, trend and residual components. (Qiu et al., 2018) proposed the structural design to extend decomposition into multivariate and probabilistic cases. Recently, CoST (Woo et al., 2022) and TIDER (LIU et al., 2022) show the disentangled representations of multivariate series could get significant performance improvement in forecasting and imputation tasks, respectively, with bonus of interpretability. However, they are not flexible enough to handle the continuous time field and observation noise. (Benavoli & Corani,

2021) propose a similar idea to directly utilize the state-space GPs with mixture kernels to estimate the seasonal-trend factors, but is restricted in univariate series.

**Bayesian imputation modeling.** Bayesian methods are widely used in time series imputation tasks for robust modeling and uncertainty quantification. Early work directly applies a single Bayesian model like Gaussian Process (Roberts et al., 2013) and energy models (Brakel et al., 2013) to model the dynamics. With deep learning boosting in the past ten years, it is popular to utilize the probabilistic modules with various deep networks, such as RNN (Mulyadi et al., 2021), VAE (Fortuin et al., 2020) and GAN (Yoon et al., 2018). Adopting score-based generative models (SGMs) is another promising direction for probabilistic imputation, which could be used as autoregressive denoising (Rasul et al., 2021), conditional diffusion (Tashiro et al., 2021), Schrödinger Bridge (Chen et al., 2023) and state-space blocks (Alcaraz & Strodthoff, 2022). However, most of the above methods are trained in the offline and patching-sequence manner, which lacks interpretability and may not fit streaming scenarios.

## 3 BACKGROUND

### 3.1 MULTIVARIATE TIME SERIES IMPUTATION

The classical multivariate time series imputation problem is formulated as follows. A $N$-step multivariate time series $\mathbf{X} = \{\mathbf{x}_1, \ldots, \mathbf{x}_N\} \in \mathbb{R}^{D \times N}$, where $\mathbf{x}_n \in \mathbb{R}^D$ is the $D$-size value at $n$-th step and $\mathbf{x}_n^d$ represents it's values at $d$-th channel. There is a mask matrix $\mathbf{M} \in \{0, 1\}^{D \times N}$, indicating whether the series value is observed or missing. The goal is to use the observed values, where $\mathbf{M}_{d,n} = 1$, to estimate the missing values $\mathbf{x}_n^d$, where $\mathbf{M}_{d,n} = 0$. In the above setting, the interval between two consecutive timestamps is assumed to be constant by default. If the timestamps are irregularly sampled and continuous, the problem becomes more challenging and the exact timestamps $\{t_1, \ldots, t_N\}$ should be considered in the imputation model. In this paper, we aimed to learn a general function $\mathbf{X}(t) : t \to \mathbb{R}^D$ to impute the missing values at any time $t \in [t_1, t_N]$.

### 3.2 GAUSSIAN PROCESS (GP) PRIOR AND ITS LTI-SDE FORMULATION

**Gaussian Process (GP)** (Rasmussen & Williams, 2006)s is a powerful Bayesian prior for functional approximation, always denoted as $f \sim \mathcal{GP}(0, \kappa(\mathbf{x}, \mathbf{x}'))$. As a non-parametric model, it's characterized by a mean function, here assumed to be zero, and a covariance function or kernel $\kappa(\mathbf{x}, \mathbf{x}')$, which is a positive definite function that measures the similarity between two inputs. The choice of the kernel is crucial for GP as it determines the types of functions the GP can model. For instance, the Matérn kernel: $\kappa_{\text{Matérn}} = \sigma^2 \frac{\left(\frac{\sqrt{2\nu}}{l}\alpha(\mathbf{x},\mathbf{x}')\right)^\nu}{\Gamma(\nu)2^{\nu-1}} K_\nu \left(\frac{\sqrt{2\nu}}{l}\alpha(\mathbf{x}, \mathbf{x}')\right)$ and periodic kernel: $\kappa_{\text{periodic}} = \sigma^2 exp\left(-2\sin^2(\pi\alpha(\mathbf{x}, \mathbf{x}')/p)/l^2\right)$ are versatile choices to model functions with non-linear and cyclical patterns, respectively. $\{\sigma^2, l, \nu, p\}$ are hyperparameters determining the variance, length-scale, smoothness, and periodicity of the function. $\alpha(\cdot, \cdot)$ is the Euclidean distance, and $K_\nu$ is the modified Bessel function, $\Gamma(\cdot)$ is the Gamma function.

Despite the flexibility and capacity, full GP is a computationally expensive model with $\mathcal{O}(n^3)$ inference cost while handliling $n$ observation data, which is not feasible in practice. To sidestep expensive kernel matrix computation, (Hartikainen & Särkkä, 2010; Särkkä, 2013) applied the spectral analysis and worked out a crucial statement: a temporal GP with a stationary kernel is equivalent to a linear time-invariant stochastic differential equation (LTI-SDE). Specifically, given $f(t) \sim \mathcal{GP}(0, \kappa(t, t'))$, we can define a vector-valued companion form: $\mathbf{z}(t) = \left(f(t), \frac{\mathrm{d}f(t)}{\mathrm{d}t}, \ldots, \frac{\mathrm{d}f^m(t)}{\mathrm{d}t}\right)^\top : t \to \mathcal{R}^{m+1}$, where $m$ is the order of the derivative. Then, the GP is equivalent to the solution of the LTI-SDE with canonical form:

$$\frac{\mathrm{d}\mathbf{z}(t)}{\mathrm{d}t} = \mathbf{F}\mathbf{z}(t) + \mathbf{L}w(t), \tag{1}$$

where $\mathbf{F}$ is a $(m+1) \times (m+1)$ matrix, $\mathbf{L}$ is a $(m+1) \times 1$ vector, and $w(t)$ is a white noise process with spectral density $q_{\mathbf{s}}$. On arbitrary collection of timestamps $\{t_1, \ldots, t_N\}$, the LTI-SDE 1 can be further discretized as the Markov model with Gaussian transition, defined as:

$$p(\mathbf{z}(t_1)) = \mathcal{N}(\mathbf{z}(t_1)|\mathbf{0}, \mathbf{P}_\infty); p(\mathbf{z}(t_{n+1})|\mathbf{z}(t_n)) = \mathcal{N}(\mathbf{z}(t_{n+1})|\mathbf{A}_n\mathbf{z}(t_n), \mathbf{Q}_n) \tag{2}$$

where $\mathbf{A}_n = \exp(\mathbf{F}\Delta_n)$, $\mathbf{Q}_n = \int_{t_n}^{t_{n+1}} \mathbf{A}_n \mathbf{L}\mathbf{L}^\top \mathbf{A}_n^\top q_\mathbf{s} \mathrm{d}t$, $\Delta_n = t_{n+1} - t_n$, and $\mathbf{P}_\infty$ is the steady-state covariance matrix of the LTI-SDE 1, which can be obtained by solving the Lyapunov equation $\mathbf{F}\mathbf{P}_\infty + \mathbf{P}_\infty \mathbf{F}^\top + \mathbf{L}\mathbf{L}^\top q_\mathbf{s} = 0$ (Lancaster & Rodman, 1995). With proper design of the observation, 2 becomes a state-space model (SSM) and can be efficiently solved with linear cost by classical methods like the Kalman filter (Kalman, 1960). After inference over $\mathbf{z}(t)$, we can easily obtain $f(t)$ by a simple projection: $f(t) = [1, 0, \ldots 0]\mathbf{z}(t)$.

We highlight that all the parameters of the LTI-SDE 1 and its discretized form 2: $\{m, \mathbf{F}, \mathbf{L}, q_\mathbf{s}, \mathbf{P}_\infty\}$ are time-invariant constant and can be derived from the given stationary kernel function. In practice, stationary kernels are a common choice for GP, which requires the kernel to be a function of the distance between two inputs. For example, the Matérn and periodic kernels are stationary kernels, and we can work out their closed-form formulas of LTI-SDE and discrete model. We omit the specific formulas here and refer the readers to the appendix.

## 4 METHOD

### 4.1 FUNCTIONAL DECOMPOSITION OF MULTIVARIATE TIME SERIES

The motivation of *BayOTIDE* is based on the fact that the different channels of real-world multivariate time series $\mathbf{X}(t)$ are always correlated, and there may exist shared temporal patterns across channels. Thus, we propose to decompose each channel of $\mathbf{X}(t)$ into the channel-wise combination weights, and groups of function factors — acting as bases of temporal patterns and shared by all channels.

Inspired by the classic seasonal-trend decomposition(STL) (Cleveland et al., 1990) and TIDER (LIU et al., 2022), we assume there are two groups of factors representing different temporal patterns. The first group of factors is supposed to capture the nonlinear and long-term patterns, and the second represents the periodic parts, namely, trends and seasonalities. Thus, we decompose the function $\mathbf{X}(t) : t \to \mathbb{R}^D$ as the weighted combination of two groups of functional factors. Specifically, assume there are $D_r$ trend factors and $D_s$ seasonality factors, then we have the following decomposition:

$$\mathbf{X}(t) = \mathbf{U}\mathbf{V}(t) = [\mathbf{U}_{\text{trend}}, \mathbf{U}_{\text{season}}] \left[ \begin{array}{c} \mathbf{V}_{\text{trend}}(t), \\ \mathbf{V}_{\text{season}}(t) \end{array} \right], \tag{3}$$

where $\mathbf{U} = [\mathbf{U}_{\text{trend}} \in \mathbb{R}^{D \times D_r}, \mathbf{U}_{\text{season}} \in \mathbb{R}^{D \times D_s}]$ are the weights of the combination. The trends factor group $\mathbf{V}_{\text{season}}(t) : t \to \mathbb{R}^{D_s}$, and seasonality factor groups $\mathbf{V}_{\text{trend}}(t) : t \to \mathbb{R}^{D_r}$ are the concatenation of independent temporal factors over each dimension:

$$\mathbf{V}_{\text{trend}}(t) = \text{concat}[\mathbf{V}_{\text{trend}}^i(t)]_{i=1\ldots D_r}, \ \mathbf{V}_{\text{season}}(t) = \text{concat}[\mathbf{V}_{\text{season}}^j(t)]_{j=1\ldots D_s}, \tag{4}$$

where $\mathbf{V}_{\text{trend}}^i(t) : t \to \mathbb{R}$ is the 1-D temporal function factor, the same with $\mathbf{V}_{\text{season}}^j(t)$.

For the imputation task, if we can estimate the $\mathbf{U}$ and $\mathbf{V}(t)$ well, we can impute the missing values of $\mathbf{X}(t)$ by $\mathbf{U}\mathbf{V}(t)$ for any $t$. As TIDER (LIU et al., 2022) proposed a low-rank decomposition similar to 3, our model can be seen as a generalization of TIDER to the continuous-time and functional field with the Bayesian framework.

### 4.2 GP PRIOR AND JOINT PROBABILITY OF OUR MODEL

We assume $\mathbf{X}(t)$ is partially observed with missing values and noise on timestamps $\{t_1, \ldots t_N\}$. We denote the observed values as $\mathbf{Y} = \{\mathbf{y}_n\}_{n=1}^N$, where $\mathbf{y}_n \in \mathbb{R}^D$, and its value at $d$-th channel is denoted as $\mathbf{y}_n^d$. $\mathbf{M} \in \{0, 1\}^{D \times N}$ is the mask matrix, where 1 for observed values and 0 for missing values. The noise level is assumed to be the same for all the channels. Thus, we set the Gaussian likelihood for the observed values as:

$$p(\mathbf{Y}|\mathbf{U}, \mathbf{V}(t), \tau) = \prod_{(d,n) \in \Omega} \mathcal{N}(\mathbf{y}_n^d \mid \mathbf{U}^d \mathbf{V}(t_n), \tau^{-1}), \tag{5}$$

where $\tau$ is the inverse of the noise level. $\Omega$ is the collection of observed values' location, namely $\Omega = \{(d, n) \mid \mathbf{M}_{d,n} = 1\}$. $\mathbf{U}^d \in \mathbb{R}^{1 \times (D_r + D_s)}$ is the $d$-th row of $\mathbf{U}$, and $\mathbf{V}(t_n) \in \mathbb{R}^{(D_r + D_s) \times 1}$ is the concatenation of $\mathbf{V}_{\text{trend}}(t_n)$ and $\mathbf{V}_{\text{season}}(t_n)$.

As $\mathbf{V}_{\text{season}}(t)$ and $\mathbf{V}_{\text{trend}}(t)$ are supposed to capture different temporal patterns, we adopt Gaussian Processes (GP) with different kernel to model them. Specifically, we use the Matérn kernel to model the trend factors, and the periodic kernel to model the seasonality factors:

$$\mathbf{V}_{\text{trend}}^i(t) \sim \mathcal{GP}(0, \kappa_{\text{Matérn}}(t, t')), \ \mathbf{V}_{\text{season}}^j(t) \sim \mathcal{GP}(0, \kappa_{\text{periodic}}(t, t')). \tag{6}$$

We further assume that each channel's weights $\mathbf{U}^d$ is with independent Gaussian priors, and assign a Gamma distribution as prior for $\tau$. Then, the joint probability model is:

$$p(\mathbf{Y}, \mathbf{V}(t), \mathbf{U}, \tau) = p(\tau) \prod_{d=1}^D p(\mathbf{U}^d) \prod_{j=1}^{D_s} \mathcal{GP}(0, \kappa_{\text{periodic}}) \prod_{i=1}^{D_r} \mathcal{GP}(0, \kappa_{\text{Matérn}}) p(\mathbf{Y}|\mathbf{U}, \mathbf{V}(t), \tau), \tag{7}$$

where $p(\tau) = \text{Gamma}(\tau \mid a_0, b_0)$, and $p(\mathbf{U}^d) = \mathcal{N}(\mathbf{U}^d \mid \mathbf{0}, \mathbf{I})$. As introduced in section 3.2, each GP prior term corresponds to a companion form $\mathbf{z}(t)$. Thus, we denote the concatenation of all factors' companion forms as $\mathbf{Z}(t)$ and $p(\mathbf{V}(t)) = p(\mathbf{Z}(t)) = P(\mathbf{Z}(t_1)) \prod_{i=1}^{N-1} P(\mathbf{Z}(t_{n+1})|\mathbf{Z}(t_n))$, which is aligned with 2. For compactness, we denote all the random variables of the model as $\Theta = \{\mathbf{U}, \mathbf{Z}(t), \tau\}$ over $t \in \{t_1, \ldots t_N\}$.

## 4.3 ONLINE INFERENCE

With the probabilistic model 7, we further propose an online inference algorithm to estimate the running posterior of $\Theta$. We denote all observations up to time $t_n$ as $\mathcal{D}_{t_n}$, i.e. $\mathcal{D}_{t_n} = \{\mathbf{y}_1, \ldots, \mathbf{y}_n\}$. When a new observation $\mathbf{y}_{n+1}$ arrives at $t_{n+1}$, we aimed to update the posterior distribution $p(\Theta \mid \mathcal{D}_{t_n} \cup \mathbf{y}_{n+1})$ without reusing the previous observations $\mathcal{D}_{t_n}$. The general principle for online inference is the incremental version of Bayes'rule, which is:

$$p(\Theta \mid \mathcal{D}_{t_n} \cup \mathbf{y}_{n+1}) \propto p(\mathbf{y}_{n+1} \mid \Theta, \mathcal{D}_{t_n}) p(\Theta \mid \mathcal{D}_{t_n}). \tag{8}$$

However, the exact posterior of $\Theta$ is not tractable. Thus, we first apply the mean-field assumption to factorize the posterior for approximate inference. Specifically, we approximate the posterior as: $p(\Theta \mid \mathcal{D}_{t_n}) \approx q(\Theta \mid \mathcal{D}_{t_n}) = q(\tau \mid \mathcal{D}_{t_n}) \prod_{d=1}^D q(\mathbf{U}^d \mid \mathcal{D}_{t_n}) q(\mathbf{Z}(t) \mid \mathcal{D}_{t_n})$ where $q(\mathbf{U}^d \mid \mathcal{D}_{t_n}) = \mathcal{N}(\mathbf{m}_n^d, \mathbf{V}_n^d)$ and $q(\tau \mid \mathcal{D}_{t_n}) = \text{Gamma}(\tau \mid a_n, b_n)$ are the approx. distributions for $\mathbf{U}$ and $\tau$ at time $t_n$ respectively. For $\mathbf{Z}(t)$, we design the approximated posterior as $q(\mathbf{Z}(t) \mid \mathcal{D}_{t_n}) = \prod_{i=1}^n q(\mathbf{Z}(t_i))$ , to align with the chain structure GP priors 2, where $q(\mathbf{Z}(t_i))$ are the concatenation of $q(\mathbf{z}(t_i)) = \mathcal{N}(\mu_i, \mathbf{S}_i)$ across all factors. $\{\{\mathbf{m}_n^d, \mathbf{V}_n^d\}, \{\mu_i, \mathbf{S}_i\}, a_n, b_n,\}$ are the variational parameters of the approximated posterior to be estimated.

With mean-field approximation, 8 is still not feasible. It's because the multiple factors and weights interweave in the likelihood, and R.H.S of 8 is unnormalized. To solve it, we propose a novel online approach to update $q(\Theta \mid \mathcal{D}_{t_n})$ with a closed form by adopting conditional Expectation Propagation(CEP) (Wang & Zhe, 2019) and chain structure of $\mathbf{Z}(t)$. Specifically, with current $q(\Theta \mid \mathcal{D}_{t_n})$ and prior $p(\Theta)$, we can approximate each likelihood term of the new-arriving observation $\mathbf{y}_{n+1}$ as several messages factors with close-form:

$$p(\mathbf{y}_{n+1}^d \mid \mathbf{U}^d, \mathbf{V}(t_n), \tau) \approx \mathcal{Z} f_{n+1}^d(\mathbf{Z}(t_{n+1})) f_{n+1}^d(\mathbf{U}_d) f_{n+1}^d(\tau), \tag{9}$$

where $\mathcal{Z}$ is the normalization constant, $f_{n+1}^d(\mathbf{U}_d) = \mathcal{N}(\mathbf{U}_d \mid \hat{\mathbf{m}}_{n+1}^d, \hat{\mathbf{V}}_{n+1}^d)$ and $f_{n+1}^d(\tau) = \text{Gamma}(\tau \mid \hat{a}_{n+1}, \hat{b}_{n+1})$, $f_{n+1}^d(\mathbf{Z}(t_{n+1})) = \text{concat}[\mathcal{N}(\hat{\mu}_i, \hat{S}_i)]$ are the approximated message factors of $\mathbf{U}_d$, $\tau$ $\mathbf{Z}(t_{n+1})$ respectively. Then, we merge all the message factors of $\mathbf{U}$ and $\tau$ and follow the variational form of 8, and will obtain the update equations:

$$q(\tau|\mathcal{D}_{t_{n+1}}) = \text{Gamma}(\tau \mid a_{n+1}, b_{n+1}) = \text{Gamma}(\tau \mid a_n, b_n) \prod_d f_{n+1}^d(\tau), \tag{10}$$

$$q(\mathbf{U}^d|\mathcal{D}_{t_{n+1}}) = \mathcal{N}(\mathbf{U}^d \mid \mathbf{m}_{n+1}^d, \mathbf{V}_{n+1}^d) = \mathcal{N}(\mathbf{U}^d \mid \mathbf{m}_n^d, \mathbf{V}_n^d) f_{n+1}^d(\mathbf{U}^d). \tag{11}$$

As the R.H.S of 10 and 11 are all in the same distribution family in the exponential family, we can obtain the closed-form update equations of $\{\{\mathbf{m}_{n+1}^d, \mathbf{V}_{n+1}^d\}, a_{n+1}, b_{n+1}\}$. The message approximation in 9 and the message merging is based on the conditional moment-matching technique, which can be done in parallel for all channels. We omit the derivation and exact formulas and refer the readers to the appendix.

---

**Algorithm 1** *BayOTIDE*

---

**Input:** observation $\mathbf{Y} = \{\mathbf{y}_n\}_{n=1}^N$ over $\{t_n\}_{n=1}^N$, $D_s$, $D_r$, the kernel hyperparameters.
Initialize the approx. posterior $q(\tau), q(\mathcal{W}), \{q(\mathbf{Z}(t_n))\}_{n=1}^N$.
**for** $t = 1$ **to** $N$ **do**
    Approximate the likelihood messages by (9) for all observed channels in parallel.
    Update the posterior of $\tau$ and $\mathbf{U}$ by (10) and (11) for all observed channels in parallel.
    Update the posterior of $\mathbf{Z}(t)$ using Kalman filter by (12).
**end for**
(optional) Run RTS smoother to obtain the full posterior of $\mathbf{Z}(t)$.
**Return:** $q(\tau), q(\mathcal{W}), \{q(\mathbf{Z}(t_n))\}_{n=1}^N$

---

Then, we present the online update of $\mathbf{Z}(t)$. With the chain structure of $q(\mathbf{Z}(t) \mid \mathcal{D}_{t_n})$ and $p(\mathbf{Z}(t))$, we found the update can be conducted sequentially. Specifically:

$$q(\mathbf{Z}(t_{n+1})) = q(\mathbf{Z}(t_n))p(\mathbf{Z}(t_{n+1})|\mathbf{Z}(t_n)) \prod_d f_{n+1}^d(\mathbf{Z}(t_{n+1})), \tag{12}$$

where $p(\mathbf{Z}(t_{n+1})|\mathbf{Z}(t_n))$ is the concatenation of all factors' transition given in (2). If we regard $\prod_d f_{n+1}^d(\mathbf{Z}(t_{n+1}))$ as the observation of the state space, 12 is the Kalman filter model (Kalman, 1960). Thus, we can obtain the closed-form update of $q(\mathbf{Z}(t)| \mid \mathcal{D}_{t_n})$, which is the running posterior $q(\mathbf{Z}_n|\mathbf{y}_{1:n})$. We can run the classical Rauch-Tung-Striebel (RTS) smoother (Rauch et al., 1965) to efficiently compute the full posterior of each state $q(\mathbf{Z}_n|\mathbf{y}_{1:N})$ from backward after going through all the timestamp.

The online algorithm is summarized in Algorithm table 1: we go through all the timestamps, approximate the message factors with moment-matching, and run Kalman filter and message merging and update sequentially. For each timestamp, we can run moment-matching and posterior update steps iteratively several times with damping trick Minka (2001a) for better approximation. The algorithm is very efficient as the message approximation 9 can be parallel for different channels, and all the updates are closed-form. The algorithm is with time cost $\mathcal{O}(N(D_s + D_r))$ and space cost $\mathcal{O}(\sum_{k=1}^{D_r+D_s} N(m_k + m_k^2) + D(D_s + D_r))$ where $N$ is the number of timestamps, $D$ is the number of channel of original time series, $D_r$, $D_s$ are number of trends and seasonality factors respectively, $m_k$ is the order of the companion form of $k$-th factor's GP prior determined by the kernel types.

### 4.4 PROBABILISTIC IMPUTATION AT ARBITRARY TIME STAMPS

With the current posterior $\{q(\mathbf{z}(t_1)) \ldots q(\mathbf{z}(t_N))\}$ over the observed timestamps, the functional and chain property of GP priors allow us to infer the prediction distribution, namely do probabilistic interpolation at arbitrary time stamps. Specifically, for a never-seen timestamp $t^\star \in (t_1, \mathbf{t}_N)$, we can identify the closest neighbor of $t^\star$ observed in training, i.e, $t_k < t^\star < t_{k+1}$, where $t_k, t_{k+1} \in \{t_1 \ldots t_N\}$. Then the predictive distribution at $t^\star$ is given by $q(\mathbf{z}(t^\star)) = \mathcal{N}(\mathbf{z}^\star \mid \mathbf{m}^\star, \mathbf{V}^\star)$, where:

$$\mathbf{V}^\star = (\mathcal{Q}_1^{-1} + \mathcal{A}_2^\top \mathcal{Q}_1^{-1} \mathcal{A}_2)^{-1}, \ \mathbf{m}^\star = \mathbf{V}^\star(\mathcal{Q}_1^{-1}\mathcal{A}_1\mathbf{m}_k + \mathcal{A}_2^\top \mathcal{Q}_2^{-1}\mathbf{m}_{k+1}), \tag{13}$$

$\mathbf{m}_k, \mathbf{m}_{k+1}$ are the predictive mean of $q(\mathbf{z}(t_k)), q(\mathbf{z}(t_{k+1}))$, $\{\mathcal{A}_1, \mathcal{A}_2, \mathcal{Q}_1, \mathcal{Q}_2\}$ are transition matrices and covariance matrices based on the forward-backward transitions $p(\mathbf{z}(t^\star)|\mathbf{z}(t_k)), p(\mathbf{z}(t_{k+1})|\mathbf{z}(t^\star))$ respectively. 13 offers continuous modeling of the time series. The derivation is given in the appendix.

## 5 EXPERIMENTS

### 5.1 SYNTHETIC DATA

We first evaluate *BayOTIDE* on a synthetic task. We first set four temporal functions and a weight matrix, defined as follows:

$$\mathbf{U} = \begin{pmatrix} 1 & 1 & -2 & -2 \\ 0.4 & 1 & 2 & -1 \\ -0.3 & 2 & 1 & -1 \\ -1 & 1 & 1 & 0.5 \end{pmatrix}, \quad \mathbf{V}(t) = \begin{pmatrix} 10t, \\ \sin(20\pi t), \\ \cos(40\pi t), \\ \sin(60\pi t) \end{pmatrix}. \tag{14}$$

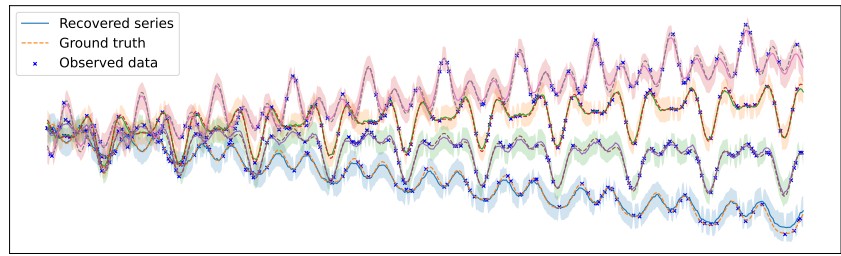

(a) Imputation results of the four-channel synthetic time series.

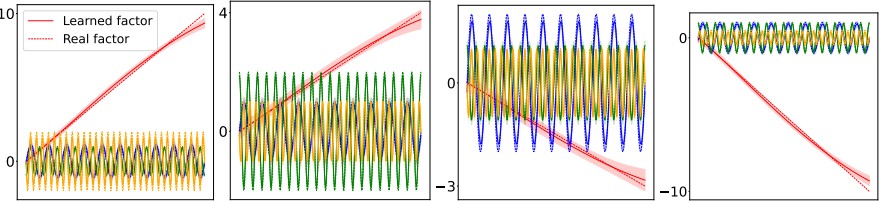

(b) Channel#1's factors (c) Channel#2's factors (d) Channel#3's factors (e) Channel#4's factors

Figure 1: (a): The multivariate time series recovered from observations. The shaded region indicates two posterior standard deviations. (b)-(e): The weighted trend-seasonality factors learned by *BayOTIDE* of each channel.

Then, the four-channel time series is defined as $\mathbf{X}(t) = \mathbf{U}\mathbf{V}(t)$, and each channel is a mixture of multiscale trend and seasonality factors. We collected 2000 data points over the 500 irregularly sampled timestamps from $[0, 1]$. We randomly set only $20\%$ of the data as observed values, and the rest as missing for evaluation. We further add Gaussian noise with a standard deviation $0.1$ to the observed data. We ues the Matérn kernel with $\nu = 3/2$ as the trend kernel and the periodic kernel with period $20\pi$ as the seasonality kernel. We set $D_r = 1, D_s = 3$. Detailed settings are in the Appendix. We highlight that evaluation could be taken on the never-seen-in-training timestamps, so we apply (13) to handle such hard cases easily.

The imputation results are shown in Figure 1a. We can see that *BayOTIDE* recovers the series well, and the estimated uncertainty is reasonable. We also show the channel-wise estimated factors in Figure 1b1c1d1e. We can see that the estimated factors are close to the real ones, which indicates that *BayOTIDE* can capture the underlying multiscale patterns of the data.

## 5.2 REAL-WORLD APPLICATIONS

**Dataset.** We evaluate *BayOTIDE* on three real-world datasets, *Traffic-Guangzhou*(Chen et al.): traffic speed records in GUangzhou with 214 channels and 500 timestamps. *Solar-Power*(https://www.nrel.gov/grid/solar-power-data.html): 137 channels and 52560 timestamps, which records the solar power generation of 137 PV plants. *Uber-Move*(https://movement.uber.com/): 7489 channels and 744 timestamps, recording the average movement of Uber cars along with the road segments in London, Jan 2020. For each dataset, we randomly sample $\{70\%, 50\%\}$ of the available data points as observations for model training, and the rest for evaluation. The data process and split strategy are aligned with TIDER (LIU et al., 2022).

**Baselines and setting.** To the best of our knowledge, there are no online algorithms for multivariate time series imputation. Thus, we set several popular deterministic and probabilistic offline approaches as baselines. The deterministic group includes: (1) *SimpleMean* (Acuna & Rodriguez, 2004), (2) *BRITS* (Cao et al., 2018), (3) *NAOMI*(Liu et al., 2019), (4) *SAITS*(Du et al., 2023), (5) *TIDER*(LIU et al., 2022). The probabilistic group includes: (1) *Multi-Task GP*(Bonilla et al., 2008), (2) *GP-VAE*(Fortuin et al., 2020), (3) *CSDI*(Tashiro et al., 2021) (4)*CSBI*(Chen et al., 2023). We also set *BayOTIDE-fix-wight* by fixing all weight values as one and *BayOTIDE-trend-only*, and only using trend factor, respectively for *BayOTIDE* . We use the released implementation provided by the authors for baselines. We partially use the results of deterministic methods reported in TIDER, as the setting is aligned. For *BayOTIDE* , we implemented it by Pytorch and finetuned the $D_s, D_r$, and the kernel parameters to obtain optimal results. Detailed information and setting of the baselines and *BayOTIDE*

| Observed-ratio=50% | Traffic-GuangZhou | | | Solar-Power | | | Uber-Move | | |
| Metrics | RMSE | MAE | CRPS | RMSE | MAE | CRPS | RMSE | MAE | CRPS |
|---|---|---|---|---|---|---|---|---|---|
| *Deterministic & Offline* | | | | | | | | | |
| SimpleMean | 9.852 | 7.791 | - | 3.213 | 2.212 | - | 5.183 | 4.129 | - |
| BRITS | 4.874 | 3.335 | - | 2.842 | 1.985 | - | 2.180 | 1.527 | - |
| NAOMI | 5.986 | 4.543 | - | 2.918 | 2.112 | - | 2.343 | 1.658 | - |
| SAITS | 4.839 | 3.391 | - | 2.791 | 1.827 | - | 1.998 | 1.453 | - |
| TIDER | 4.708 | 3.469 | - | **1.679** | 0.838 | - | 1.959 | 1.422 | - |
| *Probabilistic & Offline* | | | | | | | | | |
| Multi-Task GP | 4.887 | 3.530 | 0.092 | 2.847 | 1.706 | 0.203 | 3.625 | 2.365 | 0.121 |
| GP-VAE | 4.844 | 3.419 | 0.084 | 3.720 | 1.810 | 0.368 | 5.399 | 3.622 | 0.203 |
| CSDI | 4.813 | 3.202 | 0.076 | 2.276 | 0.804 | 0.166 | 1.982 | 1.437 | 0.072 |
| CSBI | 4.790 | 3.182 | 0.074 | 2.097 | 1.033 | 0.153 | 1.985 | 1.441 | 0.075 |
| *Probabilistic & Online* | | | | | | | | | |
| BayOTIDE-fix weight | 11.032 | 9.294 | 0.728 | 5.245 | 2.153 | 0.374 | 5.950 | 4.863 | 0.209 |
| BayOTIDE-trend only | 4.188 | 2.875 | 0.059 | 1.789 | 0.791 | 0.132 | 2.052 | 1.464 | 0.067 |
| BayOTIDE | **3.820** | **2.687** | **0.055** | 1.699 | **0.734** | **0.122** | **1.901** | **1.361** | **0.062** |

Table 2: RMSE, MAE and CRPS scores of imputation results of all methods on three datasets with observed ratio $= 50\%$. The results with observed ratio $= 70\%$ and negative log-likelihood score can be found in Appendix.

| Dataset | Traffic-GuangZhou | | Solar-Power | | Uber-Move | |
| Observed ratio | 50% | 70% | 50% | 70% | 50% | 70% |
|---|---|---|---|---|---|---|
| RMSE | 3.625 | 3.383 | 1.624 | 1.442 | 3.017 | 2.931 |
| MAE | 2.524 | 2.401 | 0.706 | 0.614 | 1.199 | 1.154 |
| CRPS | 0.051 | 0.046 | 0.121 | 0.113 | 0.311 | 0.302 |
| NLLK | 2.708 | 2.634 | 1.861 | 1.857 | 2.137 | 2.138 |

Table 3: The imputation results of *BayOTIDE* with settings of irregulate timestamps and all-channel missing on all datasets with observed ratio $= \{50\%, 70\%\}$

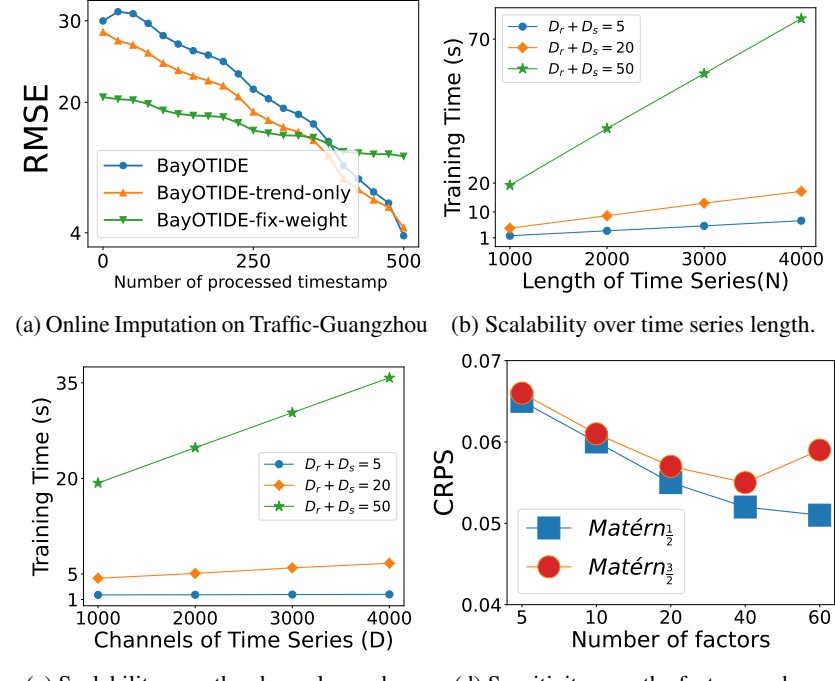

(a) Online Imputation on Traffic-Guangzhou

(b) Scalability over time series length.

(c) Scalability over the channels number.

(d) Sensitivity over the factor number

Figure 2: Online performance, scalability and sensitivity of *BayOTIDE*

are provided in the appendix. For the metrics, we use the mean absolute error (MAE) and the root mean squared error (RMSE) as the deterministic evaluation metrics for all methods. We adopt the continuous ranked probability score (CRPS) and the negative log-likelihood (NLLK), for *BayOTIDE* and all probabilistic baselines. We use 50 samples from the posterior to compute CRPS and NLLK. We repeat all the experiments 5 times and report the average results.

**Deterministic and Probabilistic performance** Table 2 shows the RMSE, MAE, and CRPS scores of imputation on three datasets with observed ratio $= 50\%$. We can see that *BayOTIDE* , an online method that only processes data once, beats the offline baselines and performs best in most cases. *TIDER* is the sub-optimal method for most cases. For probabilistic approaches, diffusion-based *CSDI* and *CSBI* obtain fair performance, but are costly in memory and need patched-sequence for training. *BayOTIDE-fix-wight* is with poor performance, which indicates that the weighted bases mechanism is effective. *BayOTIDE-trend-only* is slightly worse than *BayOTIDE* , showing the modeling of periodic factor is necessary. The results of these three scores with the observed ratio $= 70\%$ and NLLK score can be found in the appendix.

**Online Imputation** We demonstrate the online imputation performance of *BayOTIDE* on three datasets with observed ratio $50\%$. Whenever a group of observations at new timestamps have been sequentially fed into the model, we evaluate the test RMSE of the model with the updated weights and temporal factor. We compare the performance of *BayOTIDE* with the *BayOTIDE-fix-wight*. The online result on *Traffic-Guangzhou* is shown in Figure 2a. We can see that *BayOTIDE* shows the reasonable results that the evaluation error drops gradually when more timestamps are processed, meaning the model can continuously learn and improve. The performance of *BayOTIDE-fix-wight* is very poor. It indicates the trivial usage of the GP-SS model for multivariate time series imputation may not be feasible. The online results for the other two datasets are in the appendix.

**Scalability and sensitivity** We evaluate the scalability of *BayOTIDE* over data size under three settings of factor numbers: $D_r + D_s = \{5, 20, 50\}$. As for the scalability over series length $N$, We make synthetic data with channel number $D = 1000$, increase the $N$ from 1000 to 4000, and measure the training time. The result is shown in Figure 2b. Similarly, we fix the series length $N = 1000$, increase the series channel $D$ from 1000 to 4000, and then measure the training time. The result is shown in Figure 2c. As we can see, the running time of *BayOTIDE* grows linearly in both channel and length size, and the factor number determines the slope. Therefore, *BayOTIDE* enjoys the linear scalability to the data size. In addition, we further examine the sensitivity over Matérn kernel with different smoothness $\nu = \{1/2, 3/2\}$ and factor numbers on the *Traffic-Guangzhou* with observed ratio $70\%$. We vary hyperparameters and check how performance (CRPS) changes. The result is shown in Figure 2d, and we can find more factors that will result in better performance in general for both kernel types, and the performance is relatively robust. The sensitive analysis of the kernel length scale and kernel variance are left in the appendix.

**Imputation with Irregular timestamps** We further show *BayOTIDE* can work well with irregulate timestamps with functional and continuous design. We select the observations at $\{50\%, 70\%\}$ randomly sampled irregulate timestamps for training, and evaluate the model on the left never-seen-before timestamps. We highlight that most existing advanced imputation methods cannot handle this hard case well. It's because they are based on the regular-time-interval setting, which assumes there is at least one observation at every timestamp during the training and cannot do the interpolation. However, *BayOTIDE* can apply (13) and give probabilistic imputation at arbitrary continuous timestamp. Thus, we only list the results of *BayOTIDE* on three datasets in Table 3. We can see the performance is closed or even better than the standard imputation setting shown in Table 2.

CONCLUSION

We proposed *BayOTIDE* , a novel Bayesian model for online multivariate time series imputations. We decompose the multivariate time series into a temporal function basis and channel-wise weights, and apply a group of GPs to fit the temporal function basis. An efficient online inference algorithm is developed based on the SDE representation of GPs and moment-matching techniques. Results on both synthetic and real-world datasets show that *BayOTIDE* outperforms the state-of-the-art methods in terms of both imputation accuracy and uncertainty quantification.

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

APPENDIX

### .1 LTI-SDE REPRESENTATION OF GP WITH MATÉRN KERNEL AND PERIODIC KERNEL

#### .1.1 CONNECT GP WITH LTI-SDE BY SPECTRAL ANALYSIS

We take the Matérn kernel as an example to show how to connect GP with LTI-SDE. The Matérn kernel is defined as:

$$\kappa_\nu(t, t') = a \frac{(\frac{\sqrt{2\nu}}{\rho}\Delta)^\nu}{\Gamma(\nu)2^{\nu-1}} K_\nu(\frac{\sqrt{2\nu}}{\rho}\Delta) \tag{15}$$

where $\Delta = |t - t'|$, $\Gamma(\cdot)$ is the Gamma function, $a > 0$ and $\rho > 0$ are the amplitude and length-scale parameters respectively, $K_\nu$ is the modified Bessel function of the second kind, and $\nu > 0$ controls the smoothness of sample paths from the GP prior $f(t) \sim \mathcal{GP}(0, \kappa_\nu(t, t'))$.

For a stationary Matérn kernel $\kappa_\nu(t, t') = \kappa_\nu(t - t')$, the energy spectrum density of $f(t)$ can be obtained via the Wiener-Khinchin theorem by taking the Fourier transform of $\kappa_\nu(\Delta)$:

$$S(\omega) = \frac{\sigma^2}{(\alpha^2 + \omega^2)^{m+1}} \tag{16}$$

where $\omega$ is the frequency, $\alpha = \frac{\sqrt{2\nu}}{\rho}$, and we take $\nu = m + \frac{1}{2}$ for $m \in \{0, 1, 2, ...\}$.

Expanding the polynomial gives:

$$(\alpha + i\omega)^{m+1} = \sum_{k=0}^{m} c_k(i\omega)^k + (i\omega)^{m+1} \tag{17}$$

where $c_k$ are coefficients. This allows constructing an equivalent frequency domain system:

$$\sum_{k=1}^{m} c_k(i\omega)^k \widehat{f}(\omega) + (i\omega)^{m+1}\widehat{f}(\omega) = \widehat{\beta}(\omega) \tag{18}$$

where $\widehat{f}(\omega)$ and $\widehat{\beta}(\omega)$ are Fourier transforms of $f(t)$ and white noise $w(t)$ with spectral density $q_{\mathbf{s}}$ respectively.

Taking the inverse Fourier transform yields the time domain SDE:

$$\sum_{k=1}^{m} c_k \frac{d^k f}{dt^k} + \frac{d^{m+1}f}{dt^{m+1}} = w(t) \tag{19}$$

We can further construct a new state $\mathbf{z} = (f, f^{(1)}, \ldots, f^{(m)})^\top$ (where each $f^{(k)} \triangleq \mathrm{d}^k f/\mathrm{d}t^k$) and convert (19) into a linear time-invariant (LTI) SDE,

$$\frac{\mathrm{d}\mathbf{z}(t)}{\mathrm{d}t} = \mathbf{F}\mathbf{z}(t) + \mathbf{L}w(t) \tag{20}$$

where

$$\mathbf{F} = \begin{pmatrix} 0 & 1 & & \\ & \ddots & \ddots & \\ & & 0 & 1 \\ -c_0 & \cdots & -c_{m-1} & -c_m \end{pmatrix}, \quad \mathbf{L} = \begin{pmatrix} 0 \\ \vdots \\ 0 \\ 1 \end{pmatrix}.$$

The LTI-SDE is particularly useful in that its finite set of states follows a Gauss-Markov chain, namely the state-space prior. Specifically, given arbitrary $t_1 < \ldots < t_L$, we have

$$p(\mathbf{z}(t_1), \ldots, \mathbf{z}(t_L)) = p(\mathbf{z}(t_1)) \prod_{k=1}^{L-1} p(\mathbf{z}(t_{k+1})|\mathbf{z}(t_k))$$

where

$$p(\mathbf{z}(t_1)) = \mathcal{N}(\mathbf{z}(t_1)|\mathbf{0}, \mathbf{P}_\infty),$$
$$p(\mathbf{z}(t_{n+1})|\mathbf{z}(t_n)) = \mathcal{N}(\mathbf{z}(t_{n+1})|\mathbf{A}_n\mathbf{z}(t_n), \mathbf{Q}_n) \tag{21}$$

where $\mathbf{A}_n = \exp(\mathbf{F}\Delta_n)$, $\mathbf{Q}_n = \int_{t_n}^{t_{n+1}} \mathbf{A}_n \mathbf{L}\mathbf{L}^\top \mathbf{A}_n^\top q_\mathbf{s}\mathrm{d}t$, $\Delta_n = t_{n+1} - t_n$, and $\mathbf{P}_\infty$ is the steady-state covariance matrix of the LTI-SDE 1, which can be obtained by solving the Lyapunov equation $\mathbf{F}\mathbf{P}_\infty + \mathbf{P}_\infty\mathbf{F}^\top + \mathbf{L}\mathbf{L}^\top q_\mathbf{s} = 0$ (Lancaster & Rodman, 1995), as we claimed in the main paper.

Note that for other types of stationary kernel functions, such as the periodic kernels, we can approximate the inverse spectral density $1/S(\omega)$ with a polynomial of $\omega^2$ with negative roots (Solin & Särkkä, 2014), and follow the same way to construct an LTI-SDE and state-space prior.

### .1.2 THE CLOSED-FORM OF LTI-SDE AND STATE SPACE PRIOR WITH MATÉRN KERNEL AND PERIODIC KERNEL

With the canonical form of LTI-SDE (20)and state space prior(21) and above derivation, we can work out the closed-form of LTI-SDE and state space prior for Matérn kernel and periodic kernel. We present the results in the following.

For Matérn kernel with $m = 0$, indicating the smoothness is $\nu = 0 + \frac{1}{2}$, it becomes the exponential covariance function:

$$\kappa_{\exp}(\tau) = \sigma^2 \exp\left(-\frac{\tau}{\ell}\right) \tag{22}$$

Then the parameters of the LTI-SDE and state space prior are: $\{m = 0, \mathbf{F} = -1/l, \mathbf{L} = 1, q_\mathbf{s} = 2\sigma^2/l, \mathbf{P}_\infty = \sigma^2\}$

For Matérn kernel with $m = 1$, indicating the smoothness is $\nu = 1 + \frac{1}{2} = 3/2$, the kernel becomes the Matérn 3/2 covariance function:

$$\kappa_{\text{Mat.}}(\tau) = \sigma^2 \left(1 + \frac{\sqrt{3}\tau}{\ell}\right) \exp\left(-\frac{\sqrt{3}\tau}{\ell}\right) \tag{23}$$

and the parameters of the LTI-SDE and state space prior are: $m = 1$, $\mathbf{F} = \begin{pmatrix} 0 & 1 \\ -\lambda^2 & -2\lambda \end{pmatrix}$, $\mathbf{L} = \begin{pmatrix} 0 \\ 1 \end{pmatrix}$, $\mathbf{P}_\infty = \begin{pmatrix} \sigma^2 & 0 \\ 0 & \lambda^2\sigma^2 \end{pmatrix}$, $q_\mathbf{s} = 4\lambda^3\sigma^2$, where $\lambda = \sqrt{3}/\ell$.

For the periodic kernel:

$$\kappa_{\text{periodic}}(t, t') = \sigma^2 \exp\left(-\frac{2\sin^2(\pi\Delta/p)}{l^2}\right) \tag{24}$$

with preset periodicity $p$, (Solin & Särkkä, 2014) construct corresponding SDE by a sum of n two-dimensional SDE models(m=1) of the following parameters:

$$\mathbf{F}_j = \begin{pmatrix} 0 & -\frac{2\pi}{p}j \\ \frac{2\pi}{p}j & 0 \end{pmatrix}, \mathbf{L}_j = \begin{pmatrix} 1 & 0 \\ 0 & 1 \end{pmatrix} \tag{25}$$

$\mathbf{P}_{\infty,j} = q_j^2\mathbf{I}_2$, where $q_j^2 = 2\mathrm{I}_j\left(\ell^{-2}\right)/\exp\left(\ell^{-2}\right)$, for $j = 1, 2, \ldots, n$ and $q_0^2 = \mathrm{I}_0\left(\ell^{-2}\right)/\exp\left(\ell^{-2}\right)$ (Solin et al., 2016)

### .2 DERIVATIVE OF ONLINE UPDATE EQUATIONS BY CONDITIONAL MOMENT MATCHING

### .2.1 BRIEF INTRODUCTION OF EP AND CEP

The Expectation Propagation (EP) (Minka, 2001b) and Conditional EP (CEP) (Wang & Zhe, 2019) frameworks approximate complex probabilistic models with distributions in the exponential family.

Consider a model with latent variables $\boldsymbol{\theta}$ and observed data $\mathcal{D} = \{\mathbf{y}_1, \ldots, \mathbf{y}_N\}$. The joint probability is:

$$p(\boldsymbol{\theta}, \mathcal{D}) = p(\boldsymbol{\theta}) \prod_{n=1}^{N} p(\mathbf{y}_n|\boldsymbol{\theta}) \tag{26}$$

The posterior $p(\boldsymbol{\theta}|\mathcal{D})$ is usually intractable. EP approximates each term with an exponential family distribution:

$$p(y_n|\boldsymbol{\theta}) \approx c_n f_n(\boldsymbol{\theta}) \tag{27}$$
$$p(\boldsymbol{\theta}) \approx c_0 f_0(\boldsymbol{\theta}) \tag{28}$$

where $f_n(\boldsymbol{\theta}) \propto \exp(\boldsymbol{\lambda}_n^\top \boldsymbol{\phi}(\boldsymbol{\theta}))$ are in the exponential family with natural parameters $\boldsymbol{\lambda}_n$ and sufficient statistics $\boldsymbol{\phi}(\boldsymbol{\theta})$.

The joint probability is approximated by:

$$p(\boldsymbol{\theta}, \mathcal{D}) \approx f_0(\boldsymbol{\theta}) \prod_{n=1}^{N} f_n(\boldsymbol{\theta}) \cdot \text{const} \tag{29}$$

giving a tractable approximate posterior $q(\boldsymbol{\theta}) \approx p(\boldsymbol{\theta}|\mathcal{D})$.

EP optimizes the approximations $f_n$ by repeatedly:

1) Computing the calibrated distribution $q^{\backslash n}$ excluding $f_n$.

2) Constructing the tilted distribution $\widetilde{p}$ incorporating the true likelihood.

3) Projecting $\widetilde{p}$ back to the exponential family by moment matching.

4) Updating $f_n \approx \frac{q^*}{q^{\backslash n}}$ where $q^*$ is the projection.

The intractable moment matching in Step 3 is key. CEP exploits factorized $f_n = \prod_m f_{nm}(\boldsymbol{\theta}_m)$ with disjoint $\boldsymbol{\theta}_m$. It uses nested expectations:

$$\mathbb{E}_{\widetilde{p}}[\boldsymbol{\phi}(\boldsymbol{\theta}_m)] = \mathbb{E}_{\widetilde{p}(\boldsymbol{\theta}_{\backslash m})} \mathbb{E}_{\widetilde{p}(\boldsymbol{\theta}_m|\boldsymbol{\theta}_{\backslash m})}[\boldsymbol{\phi}(\boldsymbol{\theta}_m)] \tag{30}$$

The inner expectation is tractable. For the outer expectation, CEP approximates the marginal tilted distribution with the current posterior:

$$\mathbb{E}_{\widetilde{p}(\boldsymbol{\theta}_{\backslash m})}[\mathbf{g}(\boldsymbol{\theta}_{\backslash m})] \approx \mathbb{E}_{q(\boldsymbol{\theta}_{\backslash m})}[\mathbf{g}(\boldsymbol{\theta}_{\backslash m})] \tag{31}$$

If still intractable, the delta method is used to approximate the expectation with a Taylor expansion.

Once the conditional moment $\mathbf{g}(\boldsymbol{\theta}_{\backslash m})$ is obtained, CEP substitutes the expectation $\mathbb{E}_{q(\boldsymbol{\theta}_{\backslash m})}[\boldsymbol{\theta}_{\backslash m}]$ to compute the matched moment for constructing $q^*$.

### ONLINE INFERENCE UPDATE

We then applied the EP and CEP to approximate the running posterior $p(\Theta \mid \mathcal{D}_{t_n} \cup \mathbf{y}_{n+1})$. With the incremental version of Bayes'rule (8), the key is to work out the close-form factors in the likelihood approximation (9). In other words, we adopt conditional moment match techniques to handle:

$$\mathcal{N}(\mathbf{y}_{n+1}^d \mid \mathbf{U}^d \mathbf{V}(t_{n+1}), \tau^{-1}) \approx \mathcal{Z} f_{n+1}^d(\mathbf{Z}(t_{n+1})) f_{n+1}^d(\mathbf{U}_d) f_{n+1}^d(\tau) \tag{32}$$

Then we follow the standard CEP procedure to compute the conditional moment of $\{\mathbf{Z}(t_{n+1}), \mathbf{U}_d, \tau\}$ and update $f_{n+1}^d(\mathbf{U}_d) = \mathcal{N}(\mathbf{U}_d \mid \hat{\mathbf{m}}_{n+1}^d, \hat{\mathbf{V}}_{n+1}^d)$ and $f_{n+1}^d(\tau) = \text{Gamma}(\tau \mid \hat{a}_{n+1}, \hat{b}_{n+1})$, $f_{n+1}^d(\mathbf{Z}(t_{n+1})) = \text{concat}[\mathcal{N}(\hat{\mu}_i, \hat{\mathbf{S}}_i)]$.

Specifically, for $f_{n+1}^d(\tau) = \text{Gamma}(\tau \mid \hat{a}_{n+1}, \hat{b}_{n+1})$ we have:

$$\hat{a}_{n+1} = \frac{1}{2} \tag{33}$$

$$\hat{b}_{n+1} = \frac{1}{2}\mathbb{E}_q[(\mathbf{y}_{n+1}^d - \mathbf{U}^d\mathbf{V}(t_{n+1}))^2] \tag{34}$$

For $f_{n+1}^d(\mathbf{U}_d) = \mathcal{N}(\mathbf{U}_d \mid \hat{\mathbf{m}}_{n+1}^d, \hat{\mathbf{V}}_{n+1}^d)$, we have:

$$\hat{\mathbf{V}}_{n+1}^d = [\mathbb{E}_q[\tau \cdot \mathbf{Z}(t_{n+1})\mathbf{Z}^T(t_{n+1})]]^{-1} \tag{35}$$

$$\hat{\mathbf{m}}_{n+1}^d = \hat{\mathbf{V}}_{n+1}^d \cdot \mathbb{E}_q[\tau\mathbf{y}_{n+1}^d\mathbf{Z}(t_{n+1})] \tag{36}$$

For $f_{n+1}^d(\mathbf{Z}(t_{n+1})) = \text{concat}[\mathcal{N}(\hat{\mu}_i, \hat{S}_i)] = \mathcal{N}(\hat{\mu}_{\mathbf{i}}, \hat{\mathbf{S}}_i)$, we have:

$$\hat{\mathbf{S}}_i = [\mathbb{E}_q[\tau \cdot \mathbf{U}_d\mathbf{U}_d^T]]^{-1} \tag{37}$$

$$\hat{\mu}_{\mathbf{i}} = \hat{\mathbf{S}}_i \cdot \mathbb{E}_q[\tau\mathbf{y}_{n+1}^d\mathbf{U}_d] \tag{38}$$

All the expectation is taken over the current approximated posterior $q(\Theta \mid \mathcal{D}_{t_n})$.

With these message factors from the new-arriving likelihood, the online update is easy. We follow the (10), (11) and (12) to merge the factors and obtain the closed-form online update for the global posterior.

## .3 DERIVATION OF THE PROBABILISTIC IMPUTATION AT ARBITRARY TIME STAMPS

Consider a general state space model, which includes a sequence of states $\mathbf{x}_1, \ldots, \mathbf{x}_M$ and the observed data $\mathcal{D}$. The states are at time $t_1, \ldots, t_M$ respectively. The key of the state space model is that the prior of the states is a Markov chain. The joint probability has the following form,

$$p(\mathbf{x}_1, \ldots, \mathbf{x}_M, \mathcal{D}) = p(\mathbf{x}_1) \prod_{j=1}^{M-1} p(\mathbf{x}_{j+1}|\mathbf{x}_j) \cdot p(\mathcal{D}|\mathbf{x}_1, \ldots, \mathbf{x}_M). \tag{39}$$

Note that here we do not assume the data likelihood is factorized over each state, like those typically used in Kalman filtering. In our point process model, the likelihood often couples multiple states together.

Suppose we have run some posterior inference to obtain the posterior of these states $q(\mathbf{x}_1, \ldots, \mathbf{x}_M)$, and we can easily pick up the marginal posterior of each state and each pair of the states. Now we want to calculate the posterior distribution of the state at time $t^*$ such that $t_m < t^* < t_{m+1}$. Denote the corresponding state by $\mathbf{x}^*$, our goal is to compute $p(\mathbf{x}^*|\mathcal{D})$. To do so, we consider incorporating $\mathbf{x}^*$ in the joint probability (39),

$$p(\mathbf{x}_1, \ldots, \mathbf{x}_m, \mathbf{x}^*, \mathbf{x}_{m+1}, \ldots, \mathbf{x}_M, \mathcal{D})$$

$$= p(\mathbf{x}_1) \prod_{j=1}^{m-1} p(\mathbf{x}_{j+1}|\mathbf{x}_j) \cdot p(\mathbf{x}^*|\mathbf{x}_m)p(\mathbf{x}_{m+1}|\mathbf{x}^*) \cdot \prod_{j=m+1}^{M} p(\mathbf{x}_{j+1}|\mathbf{x}_j) \cdot p(\mathcal{D}|\mathbf{x}_1, \ldots, \mathbf{x}_M). \tag{40}$$

Now, we marginalize out $\mathbf{x}_{1:M\setminus\{m,m+1\}} = \{\mathbf{x}_1, \ldots, \mathbf{x}_{m-1}, \mathbf{x}_{m+2}, \ldots, \mathbf{x}_M\}$. Note that since $\mathbf{x}^*$ does not appear in the likelihood, we can take it out from the integral,

$$p(\mathbf{x}_m, \mathbf{x}_{m+1}, \mathbf{x}^*, \mathcal{D})$$

$$= \int p(\mathbf{x}_1) \prod_{j=1}^{m-1} p(\mathbf{x}_{j+1}|\mathbf{x}_j) \prod_{j=m+1}^{M} p(\mathbf{x}_{j+1}|\mathbf{x}_j) \cdot p(\mathcal{D}|\mathbf{x}_1, \ldots, \mathbf{x}_M)\mathrm{d}\mathbf{x}_{1:M\setminus\{m,m+1\}}$$

$$\cdot p(\mathbf{x}^*|\mathbf{x}_m)p(\mathbf{x}_{m+1}|\mathbf{x}^*)$$

$$= \frac{p(\mathbf{x}_m, \mathbf{x}_{m+1}, \mathcal{D})p(\mathbf{x}^*|\mathbf{x}_m)p(\mathbf{x}_{m+1}|\mathbf{x}^*)}{p(\mathbf{x}_{m+1}|\mathbf{x}_m)}. \tag{41}$$

Therefore, we have

$$p(\mathbf{x}_m, \mathbf{x}_{m+1}, \mathbf{x}^* | \mathcal{D}) \propto p(\mathbf{x}_m, \mathbf{x}_{m+1} | \mathcal{D}) p(\mathbf{x}^* | \mathbf{x}_m) p(\mathbf{x}_{m+1} | \mathbf{x}^*). \tag{42}$$

Suppose we are able to obtain $p(\mathbf{x}_m, \mathbf{x}_{m+1} | \mathcal{D}) \approx q(\mathbf{x}_m, \mathbf{x}_{m+1})$. We now need to obtain the posterior of $\mathbf{x}^*$. In the LTI SDE model, we know that the state transition is a Gaussian jump. Let us denote

$$p(\mathbf{x}^* | \mathbf{x}_m) = \mathcal{N}(\mathbf{x}^* | \mathbf{A}_1 \mathbf{x}_m, \mathbf{Q}_1), \quad p(\mathbf{x}_{m+1} | \mathbf{x}_*) = \mathcal{N}(\mathbf{x}_{m+1} | \mathbf{A}_2 \mathbf{x}^*, \mathbf{Q}_2).$$

We can simply merge the natural parameters of the two Gaussian and obtain

$$p(\mathbf{x}_m, \mathbf{x}_{m+1}, \mathbf{x}^* | \mathcal{D}) = p(\mathbf{x}_m, \mathbf{x}_{m+1} | \mathcal{D}) \mathcal{N}(\mathbf{x}^* | \boldsymbol{\mu}^*, \boldsymbol{\Sigma}^*), \tag{43}$$

where

$$(\boldsymbol{\Sigma}^*)^{-1} = \mathbf{Q}_1^{-1} + \mathbf{A}_2^\top \mathbf{Q}_2^{-1} \mathbf{A}_2,$$
$$(\boldsymbol{\Sigma}^*)^{-1} \boldsymbol{\mu}^* = \mathbf{Q}_1^{-1} \mathbf{A}_1 \mathbf{x}_m + \mathbf{A}_2^\top \mathbf{Q}_2^{-1} \mathbf{x}_{m+1}. \tag{44}$$

## .4    DETAILED INFORMATION OF EXPERIMENTS SETTING

We provide detailed information of the baselines : (1) *SimpleMean* (Acuna & Rodriguez, 2004), impute with column-wise mean values. (2) *BRITS* (Cao et al., 2018), the RNN-based model for imputation with time decay (3) *NAOMI*(Liu et al., 2019), a Bidirectional RNN model build with adversarial training (4) *SAITS*(Du et al., 2023), a transformer-based model which adopts the self-attention mechanism. (5) *TIDER*(LIU et al., 2022). State of art deterministic imputation model based on disentangled temporal representations.

The probabilistic group includes: (1) *Multi-Task GP*(Bonilla et al., 2008), the classical multi-output Gaussian process model (2) *GP-VAE*(Fortuin et al., 2020), a deep generative model which combines Gaussian Processes(GP) and variational autoencoder(VAE) for imputation (3) *CSDI*(Tashiro et al., 2021) Famous probabilistic approach which apply conditional diffusion model to capture the temporal dependency. (4)*CSBI* Advanced diffusion method that models the imputation task as a conditional Schrödinger Bridge(SB)(Chen et al., 2023). We also set *BayOTIDE-fix-wight* by fixing all weight values as one and *BayOTIDE-trend-only*, and only using trend factor, respectively for *BayOTIDE* .

We use the released implementation provided by the authors for baselines. We partially use the results of deterministic methods reported in TIDER, as the setting is aligned. To avoid the out-of-memory problem of diffusion-based and deep-based baselines, we split the whole sequence into small patches and subsample the channels for those methods, following the setting of the original paper.

For *BayOTIDE* , we implemented it by Pytorch and adopted the whole sequence to train. For the CEP step at each timestamp in *BayOTIDE* , we use the *damping trick* (Minka, 2001a) in several inner epochs to avoid numerical instability. For the regular imputation, we run the RTS smoother for the final imputation results. For the online imputation, we use the online update equations to obtain the imputation results and run RTS smoother at every evaluation timestamp.

We finetune the $D_s$, $D_r$ and the other parameters to obtain optimal results. The setting of hyperparameters is listed in Table 4 for the real-world datasets. The hyperparameters for synthetic datasets is $\{D_r = 1, D_s = 3, \nu = 3/2\}$, Trend factor lengthscale $= 3$, Trend factor variance $= 3$, Seasonal factor frequency $= 20\pi$, Seasonal factor lengthscale $= 0.1$, Damping epochs $= 5$.

## .5    MORE EXPERIMENTAL RESULTS

The imputation results of *BayOTIDE* and baselines with  on three datasets with observed ratio $= 70\%$ are shown in table 6. The NLLK scores of probabilistic imputation approaches across all datasets with different observed ratios are shown in table 5. We can see that *BayOTIDE* , an online method that only processes data once, beats the offline baselines and achieves the best performance in all cases.

For the online imputation, the results on the *Solar-Power* and *Uber-Move* is shown in Figure 4a and Figure 4b.

For the sensitive analysis, we examine the sensitivity over Matérn kernel with different smoothness $\nu = \{1/2, 3/2\}$, lengthscale and variance on *Traffic-Guangzhou* with observed ratio $70\%$. We vary

| Observed-ratio=50% | Traffic-GuangZhou | Solar-Power | Uber-Move |
|---|---|---|---|
| Number of trend factor $D_r$ | 30 | 50 | 30 |
| Number of seasonality factor $D_s$ | 10 | 5 | 5 |
| Trend factor smoothness ($\nu$) | $\frac{1}{2}$ | $\frac{3}{2}$ | $\frac{3}{2}$ |
| Trend factor lengthscale | 0.1 | 0.001 | 0.1 |
| Trend factor variance | 1.0 | 1.0 | 1.0 |
| Seasonal factor frequency ($2\pi p$) | 15 | 10 | 15 |
| Seasonal factor lengthscale | 0.05 | 0.5 | 0.05 |
| Damping epochs | 5 | 2 | 5 |
| Observed-ratio=70% | Traffic-GuangZhou | Solar-Power | Uber-Move |
| Number of trend factor $D_r$ | 30 | 50 | 30 |
| Number of seasonality factor $D_s$ | 10 | 5 | 5 |
| Trend factor smoothness ($\nu$) | $\frac{1}{2}$ | $\frac{1}{2}$ | $\frac{1}{2}$ |
| Trend factor lengthscale | 0.1 | 0.0005 | 0.1 |
| Trend factor variance | 1.0 | 1.0 | 1.0 |
| Seasonal factor frequency ($2\pi p$) | 15 | 100 | 15 |
| Seasonal factor lengthscale | 0.05 | 0.5 | 0.05 |
| Damping epochs | 5 | 2 | 5 |

Table 4: The hyperparameter setting of *BayOTIDE* for the imputation task.

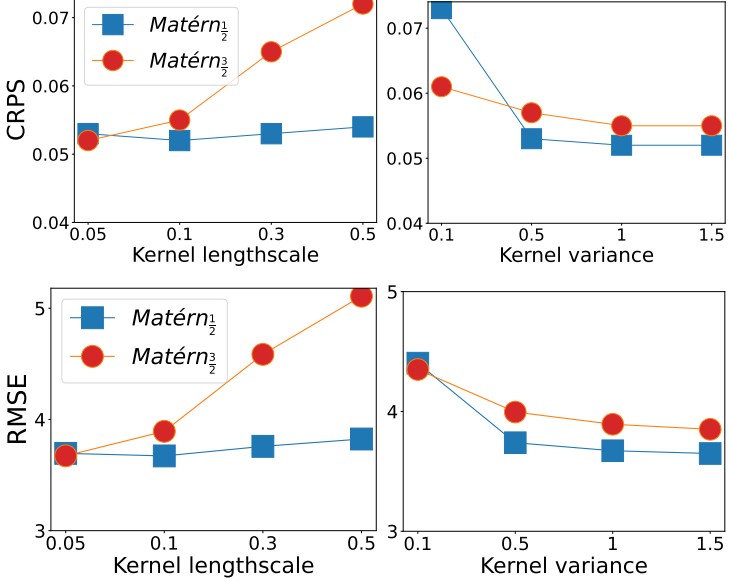

Figure 3: Sensitivity analysis of *BayOTIDE* over kernel hyperparameters.

hyperparameters and check how both deterministic and probabilistic performance (RMSE and CRPS) change. The result is shown in Figure 3, and we can find that the performance is relatively stable over different hyperparameters for Matérn kernel with smoothness $\nu = \{1/2\}$. For the Matérn kernel with smoothness $\nu = \{3/2\}$, the performance is more sensitive to the hyperparameters.

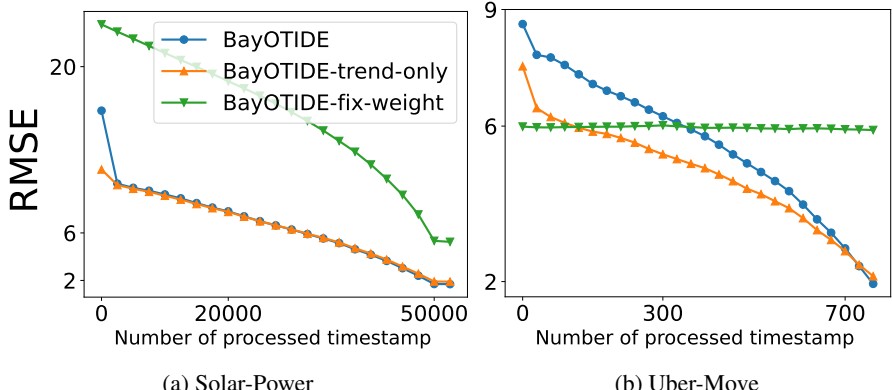

Figure 4: Online imputation results on *Solar-Power* and *Uber-Move*.

| Dataset | Traffic-GuangZhou | | Solar-Power | | Uber-Move | |
|---|---|---|---|---|---|---|
| Observed ratio | 50% | 70% | 50% | 70% | 50% | 70% |
| *Probabilistic & Offline* | | | | | | |
| Multi-Task GP | 7.339 | 6.921 | 4.921 | 4.292 | 4.426 | 4.027 |
| GP-VAE | 5.353 | 4.691 | 6.921 | 6.006 | 7.323 | 5.827 |
| CSDI | 3.942 | 3.518 | 3.433 | 2.921 | 2.415 | 2.322 |
| CSBI | 3.912 | 3.527 | 3.537 | 3.016 | 2.424 | 2.331 |
| *Probabilistic & Online* | | | | | | |
| BayOTIDE-fix weight | 10.239 | 8.905 | 4.116 | 4.093 | 3.249 | 3.252 |
| BayOTIDE-trend only | **2.897** | **2.852** | 1.944 | 1.878 | 2.169 | 2.146 |
| BayOTIDE | 3.244 | 3.078 | **1.885** | **1.852** | **2.167** | **2.100** |

Table 5: The negative log-likelihood score (NLLK) of all probabilistic imputation methods on all datasets with observed ratio $= \{50\%, 70\%\}$

| *Observed-ratio=70%* | *Traffic-GuangZhou* | | | *Solar-Power* | | | *UberLondon* | | |
|---|---|---|---|---|---|---|---|---|---|
| Metrics | RMSE | MAE | CRPS | RMSE | MAE | CRPS | RMSE | MAE | CRPS |
| *Deterministic & Offline* | | | | | | | | | |
| SimpleMean | 10.141 | 8.132 | - | 3.156 | 2.319 | - | 5.323 | 4.256 | - |
| BRITS | 4.416 | 3.003 | - | 2.617 | 1.861 | - | 2.154 | 1.488 | - |
| NAOMI | 5.173 | 4.013 | - | 2.702 | 2.003 | - | 2.139 | 1.423 | - |
| SAITS | 4.407 | 3.025 | - | 2.359 | 1.575 | - | 1.893 | 1.366 | - |
| TIDER | 4.168 | 3.098 | - | 1.676 | 0.874 | - | 1.867 | 1.354 | - |
| *Probabilistic & Offline* | | | | | | | | | |
| Multi-Task GP | 4.471 | 3.223 | 0.082 | 2.618 | 1.418 | 0.189 | 3.159 | 2.126 | 0.108 |
| GP-VAE | 4.373 | 3.156 | 0.075 | 3.561 | 1.723 | 0.331 | 3.133 | 2.005 | 0.625 |
| CSDI | 4.301 | 2.991 | 0.069 | 2.132 | 1.045 | 0.153 | 1.886 | 1.361 | 0.068 |
| CSBI | 4.201 | 2.955 | 0.064 | 1.987 | 0.926 | 0.138 | 1.899 | 1.353 | 0.070 |
| *Probabilistic & Online* | | | | | | | | | |
| BayOTIDE-fix weight | 13.319 | 9.29 | 0.677 | 5.238 | 2.026 | 0.388 | 5.889 | 4.849 | 0.208 |
| BayOTIDE-trend only | 4.002 | 2.759 | 0.056 | 1.651 | 0.712 | 0.124 | 2.015 | 1.438 | 0.065 |
| BayOTIDE | **3.724** | **2.611** | **0.053** | **1.621** | **0.709** | **0.116** | **1.832** | **1.323** | **0.061** |

Table 6: RMSE, MAE and CRPS scores of imputation results of all methods on three datasets with observed ratio $= 70\%$.

