# OpenReview forum: "BayOTIDE: Bayesian Online Multivariate Time series Imputation with functional decomposition"
_ICLR.cc/2024/Conference — Submitted to ICLR 2024_

### Official Review · Reviewer_XZJX · 2023-10-23

**Soundness:** 3 good
**Presentation:** 2 fair
**Contribution:** 3 good
**Rating:** 5
**Confidence:** 3

**Summary:**

This paper introduces BayOTIDE (Bayesian Online Multivariate Time series Imputation with functional decomposition), a novel approach for handling missing data in multivariate time series. BayOTIDE views multivariate time series as a combination of various low-rank temporal factors with distinct patterns. A group of Gaussian Processes with different kernels is utilized as functional priors. To enhance computational efficiency, the GPs are transformed into a state-space prior using an equivalent stochastic differential equation, enabling the development of a scalable online inference algorithm. One of the key advantages of the proposed method is its ability to perform imputation over arbitrary time points in the time series.

**Strengths:**

The strengths include:
1. Representing multivariate time series data through low-rank temporal factors with distinct patterns for improved insight.
2. Robust handling of missing data using probabilistic reasoning and uncertainty quantification.
3. Efficient online inference through a conversion of Gaussian Processes into a state-space prior.
4. Suitable for imputing data at arbitrary time points.

**Weaknesses:**

I am not an expert in the field of time series imputation, and therefore, I am unable to assess the novelty of this work within the imputation domain. However, I have a strong background in Bayesian frameworks and Gaussian processes. From a methodological perspective, key techniques employed in this work, such as the conversion of Gaussian Processes into a state-space model and the use of conditional Expectation Propagation for posterior approximation, are all existing methods. The methodological aspect of the paper does not introduce significant innovations. Nevertheless, it is possible that these methods could hold value within the imputation field.

Furthermore, the notation used in this paper is not ideal. As a common practice, scalars are typically represented using lowercase letters, vectors using bold lowercase letters, and matrices using bold uppercase letters. The notation employed in the paper is confusing. For instance, in Equation (1), F (a matrix) and L (a vector) are presented in the same format. Additionally, $\omega(t)$ (a scalar) in Equation (1) and $\mathbf{y}_n^d$ (a scalar) in Equation (5) are both scalars but are denoted differently. Similar issues can be found with various other symbols such as $U$, $V$, and so forth. These concerns are not exhaustively listed here.

**Questions:**

Why use Matern kernel to model the trend factors, not other kernels? Any explanation?

I understand that the conversion of Gaussian Processes into a state-space model is a computationally efficient approach to bypass the costly kernel matrix computation and facilitates the derivation of subsequent online inference. There are also other techniques based on low-rank approximations to reduce the computational complexity of GPs. Is it possible to incorporate such methods into your framework?

Recommendation: Move "We highlight that all the parameters of the LTI-SDE....can be derived from the given stationary kernel function." under equation (2). This will help readers gain a clearer understanding of how the parameters of LTI-SDE are obtained.

---

> ### Author Response · Authors · 2023-11-21
>
> We thank the reviewer for the careful review and constructive suggestions. We address the comments below:C: comments; R: response.
>
> > C1:Novelty in Bayesian learning side and time series imputation side
>
> R1: Good point! Please refer to sections **Novelty and contribution on time series imputation side** and  **Novelty and contribution on Bayesian learning side** in the response overview, which at the top-level post in this forum.
>
> > C2: Notation polish and remove inaccurate statements
>
> R2: We thank the reviewer for the careful reading and pointing out the inaccurate statements. We will polish the notation and remove the inaccurate statements in the revised version.
>
> > C3: Why matern kernel?
>
> R3: The state-space GP we used requires the kernel to be a stationary covariance function. The Matern kernel is a strong and well-known stationary covariance function, which is widely used in the GP literature and has an elegant closed form for the corresponding state-space representation(Sections 1.1 and 1.2 in the appendix). We can further adjust the smoothness parameter $\nu$ to control the smoothness of the kernel to match the different series.
>  It can be further reduced to the widely-used RBF kernel when the smoothness parameter $\nu$ is set to $1/2$.
>
>  Thus, considering the capacity and flexibility of the Matern kernel, and the elegant closed form of the corresponding state-space representation
>  we choose the Matern kernel to model the non-linear trend components in **BayOTIDE**.
>
> > C4: "Is it possible to incorporate other low-rank GP methods into your framework?"
>
> R4: Very good point! Utilizing other low-rank GP methods, like the classical sparse GP with inducing points, is feasible. We just need to run SVI to minimize the ELBO with respect to joint probability (eq 7). However, compared with state-space GP,
>  we claim the sparse GP may not be ideal for the following reasons:
>
>  - The space GP with inducing points actually is the low-rank approximation of the covariance matrix, which is could be low quality when the inducing points are not well-selected. On the contrary, the state-space GP is exactly equivalent to the original GP, which fully preserves the capacity of GPs.
>
> - The ELBO-based SVI for sparse GP is not able to handle online learning well. On the contrary, the state-space GP, formulated as a chain-structured prior, is able to handle the streaming inference with KF. It's a great advantage for the time series task.

---

> > ### Author Response · Authors · 2023-11-23
> >
> > Dear Reviewer XZJX,
> >
> > Since the End of author/reviewer discussions is just in one day, may we know if our response addresses your main concerns? If so, we kindly ask for your reconsideration of the score. Should you have any further advice on the paper and/or our rebuttal, please let us know and we will be more than happy to engage in more discussion and paper improvements.
> >
> > Thank you so much for devoting time to improving our paper!

---

> > > ### Comment · Reviewer_XZJX · 2023-12-02
> > > **Thanks for the authors' reply**
> > >
> > > Thanks for the authors' reply addressing some of my concerns. Considering the opinions of other reviewers, I still believe that the paper is not presented and evaluated very well. Therefore, I will maintain my original score.

---

### Official Review · Reviewer_PJoe · 2023-10-29

**Soundness:** 3 good
**Presentation:** 2 fair
**Contribution:** 2 fair
**Rating:** 5
**Confidence:** 3

**Summary:**

The paper proposes a Gaussian processes-based method for online multivariate time series imputation. By considering a Linear time-invariant stochastic differential equation, a solution to which is a Gaussian process, and representing it as a Markov process, the paper aims to impute missing values at arbitrary time stamps. Furthermore, the model decomposes time series into multiple channels to account for factors such as trend and seasonality. The resulting approach is capable of providing probabilistic missing data imputation in online streaming tasks.

**Strengths:**

Overall, the idea of the paper is appealing; in particular, the continuous modelling with developing methods in Neural ODEs/GPs seems a natural direction to consider. From a methodological point of view, the method is able to provide missing data imputation ability in very relevant realistic scenarios (online, continuous setting) which is definitely a notable strength.

**Weaknesses:**

The evaluation approach remains weak: I am not sure the considered data sets are not varied enough, as missing data patterns considered in the paper are quite limited. At this point, I cannot give a rating above 2 for contribution because of limited evaluation (i.e., it is unclear how well the proposed approach performs in a more general settings). The same holds for the general rating.

**Questions:**

-- Only 50% and 70% of observed ratios are considered in the simulation results. To evaluate how the proposed method compares in more general to other benchmarks, it is important to consider various benchmarks: 90%, 80%, 70%, 60%, 50%. Is the approach beneficial at all levels of missing values or do the benefits come only at a certain level?

-- Only missingness at random is considered. GP-VAE paper considers the following mechanisms: random, spatial, 2 temporal and missing, not at random.  While spatial would not be relevant here I presume, a more relevant multivariate time series mechanisms of missingness can be considered.

-- The approach considers trend and seasonality explicitly. I am not sure any of the compared benchmarks explicitly consider these channels of the time series. Therefore, I would at least include a standard multi-output GP framework with linear + periodic kernels or a spectral kernel (the implementation of the latter should be available in gpytorch). In particular, on the example represented in Figure 1 I would expect these standard methods to perform well.

-- Results in Table 2 appear over a single run; a more extensive Monte Carlo study should be considered with corresponding standard deviations in the results.

---

> ### Author Response · Authors · 2023-11-21
>
> Thanks for your comments. Here are our responses. C: comments; R: response.
>
> > C1: More missing rate setting
>
> R1: We agree that more missing rate settings will help to better evaluate the performance of the proposed method. However, we argue that the main contribution of **BayOTIDE**, the first Bayesian online imputation model, is mainly on the novel methodological and theoretical aspects, instead of the performance gain over large-scale datasets under different missing rate settings. Similar method-driven work in this area, like **CSDI** [ Tashiro, Neurips 2021],  **CSBI** [Chen, ICML 2022], and **TIDER** [Liu, ICRL 2023] all only evaluate the performance under no-more-three missing rate settings.
>
> Thus, we only evaluate the performance under two missing rate settings in the paper. We will add statements to clarify this point in the revised version.
>
> > C2: "Only missingness at random is considered."
>
> R2: Good point. We clarify that we actually evaluate the performance under **two** missing pattern settings.
>
> - The first setting is the **random missing pattern**, which is the most common missing pattern in real-world applications. Table 2 in the paper shows the performance under this missing setting.
>
>  - The second setting is the **all-channel / block-wise missing**, which is similar to the *temporal missing* pattern in the **GP-VAE** paper. We actually use the setting at **Imputation with Irregular timestamps** parts(last section of section 5). During the training, the observations at the irregular timestamps are all masked out. The results are shown in the table 3 and 5.
>
>
> >C3: Multi-output GP framework with linear + periodic kernels as baselines.
>
> R3: Good point. We do agree that the multi-output GP with linear + periodic kernels can work well in the simulation study. However, we argue that even with linear + periodic kernels,**multi-output GP can not handle the real-world long-time series. It's because the multi-output GP is a full GP, which has $O(T^2N^2)$ space cost and $O(T^3N^3)$ time cost**. where $T$ is channel number and $N$ is the number of timestamps. Thus, it's not scalable to the real-world long time series.
>
>  To make the multi-output GP work on the real-world long time series, we have to **patch the long time series into short segments**, and then train the multi-output GP on each segment (**The segment size is $30 \times 30$ in our experiments**). The patching makes the kernel can not capture any long-term dependency, like seasonal dependency, and the performance will be similar or worse than the multi-output GP with RBF kernels. To verify this, we compare the performance of multi-output GP with RBF kernels and multi-output GP with linear + periodic kernels on the *traffic-guangzhou* dataset. The results are shown in the following table:
>
> | Observed-ratio=70% | RMSE | MAE | CRPS |
> | ------ | ---- | --- | --- |
> | **BayOTIDE** | 3.724 | 2.611 |  0.053 |
> | **MT-GP-RBF** | 4.471 | 3.223 |  0.082  |
> | **MT-GP-linear+periodic** | 8.673 | 7.224 |  0.252 |
>
> > C4: MC-study on evaluation
>
> R4: We clarify that we actually use the MC-study on evaluation. We state at the end of the *Baseline and setting* paragraph in section 5.2 that "We repeat all the experiments 5 times and report the average results.". Due to the space limit, we do not report the variance of the results.

---

> > ### Author Response · Authors · 2023-11-21
> > **Response to Reviewer PJoe (before the end of rebuttal)**
> >
> > Dear Reviewer PJoe,
> >
> > Thank you so much for devoting time to improving our paper!
> >
> > Since the End of author/reviewer discussions is just in one day, may we know if our response addresses your main concerns? If so, we kindly ask for your reconsideration of the score.
> >
> > Should you have any further advice on the paper and/or our rebuttal, please let us know and we will be more than happy to engage in more discussion and paper improvements.

---

> > ### Comment · Reviewer_PJoe · 2023-11-22
> >
> > I thank the authors for the reply and for addressing some of the raised points. I believe the method is promising but not yet well presented and evaluated. Based on the latest state of the paper (I believe no revision was uploaded) and replies of the authors, I am raising my score but cannot recommend acceptance.

---

> > > ### Author Response · Authors · 2023-11-23
> > >
> > > We thank you so much for reviewing work and valuable feedback!

---

### Official Review · Reviewer_SiJc · 2023-10-31

**Soundness:** 3 good
**Presentation:** 2 fair
**Contribution:** 3 good
**Rating:** 5
**Confidence:** 3

**Summary:**

This paper proposes BAYOTIDE, a novel Bayesian model for online multivariate time series data imputation. BAYOTIDE models the data as a weighted sum of temporal factors governed by Gaussian processes. Here, the Gaussian processes can be discretized on a random collection of time steps as a Markov model with Gaussian transition. This enables the model to deal with irregularly sampled data. By viewing the model as a state-space model, an online inference procedure is derived using Kalman filtering. Thus, the model can handle missing data. Experiments on real and synthetic data show the competitiveness of the model.

**Strengths:**

- The paper tackles the problem imputing missing values of an irregularly sampled time series data. The setting is very practical. A lot of existing methods only consider regularly sampled data, and it is non-trivial for them to handle irregular data.
- This paper extends the idea of TIDER and put it in a novel framework combining Gaussian process and state-space model. This allows the model to perform 1. online imputation 2. uncertainty quantification on missing values and 3. handle irregular data
- Analysis on the complexity of running cost is given
- Experiments on real and synthetic data show the competitiveness of the proposed method

**Weaknesses:**

- There are typos and indentation issues in the paper
- Although there are no existing online multivariate imputation model, the comparison with online univariate & probabilistic imputation models, e.g. state-space model with Kalman filtering, can be included in the experiment
- RTS smoother is listed in Algorithm 1 as an option to compute the full posterior. However, it seems that the formula is not given in the paper or the appendix

**Questions:**

- The main advantage of considering multivariate time series is that the correlation between dimensions can be captured. It seems that all the evaluation metrics are univariate. I suggest the authors to also include multivariate metrics (e.g., energy score [1] and sum CRPS [2]) in the experiments to evaluate if the propose method can better capture the correlations than baselines
- The model size (e.g., number of parameters) is not reported in the experimental results. It is recommended to also report model size. The proposed model seems to be outperforming, but could it be because that it is using a larger model?

[1] Gneiting, T., & Raftery, A. E. (2007). Strictly proper scoring rules, prediction, and estimation. Journal of the American statistical Association, 102(477), 359-378.
[2] Kan, K., Aubet, F. X., Januschowski, T., Park, Y., Benidis, K., Ruthotto, L., & Gasthaus, J. (2022, May). Multivariate quantile function forecaster. In International Conference on Artificial Intelligence and Statistics (pp. 10603-10621). PMLR.

---

> ### Author Response · Authors · 2023-11-21
>
> We thank the reviewer for the careful review and constructive suggestions.We address the comments below:(C: comments; R: response.)
>
> > C1: Typos and presentation.
>
> R1: Thanks for the suggestion! We will polish the presentation quality in the revised version.
>
> > C2: Comparison with online univariate & probabilistic imputation models （state-space model with Kalman filtering）
>
> R2: Good suggestion! We add the comparison with a single state-space model (**single-SS**) with Kalman filtering on *traffic-guangzhou* The results are shown in the following table:
>
> | Observed-ratio=70% | RMSE | MAE | CRPS |
> | ------ | ---- | --- | --- |
> | **BayOTIDE** | 3.724 | 2.611 |  0.053 |
> | **single-SS** | 5.152 | 4.691 |  0.132 |
>
> | Observed-ratio=50% | RMSE | MAE | CRPS |
> | ------ | ---- | --- | --- |
> | **BayOTIDE** | 3.820 | 2.687 |  0.055 |
> | **single-SS**| 5.558 | 4.821 |  0.173 |
>
> It's clear that the **single-SS** model is much worse than the proposed method. The reason is that the **single-SS** model is not able to capture the cross-channel dependency, which is crucial for multivariate time series imputation.
>
> > C3: Add multivariate metrics (e.g., energy score [1] and sum CRPS [2])
>
> R3: We do agree that the multivariate metrics are more informative than univariate metrics, but we clarify that the *energy score*[1] will reduce to the *CRPS* ($\beta=m=1$) and *RMSE* ($\beta=2$), which are already included in the paper. For the *sum CRPS* used in [2], it seems to be the sum over the CRPS overall evaluation points. As we report the mean of the CRPS overall evaluation points, the *sum CRPS* will have the same effect as the *CRPS* in our case. However, we do agree that multivariate metrics may help to better understand the performance under multivariate setting and will investigate it in future work.
>
>
> [1] Gneiting, Tilmann, and Adrian E. Raftery. "Strictly proper scoring rules, prediction, and estimation." Journal of the American Statistical Association 102.477 (2007): 359-378.
>
>
> [2] Kan, K., Aubet, F. X., Januschowski, T., Park, Y., Benidis, K., Ruthotto, L., & Gasthaus, J. (2022, May). Multivariate quantile function forecaster. In International Conference on Artificial Intelligence and Statistics， 2022
>
> > C4: Add model size discussion
>
> R4: Good point! We take the *traffic-guangzhou* dataset(size: 213 channels, 500 timestamps) as an example, and show the model size comparison of *BayOTIDE* and *CSDI*[1]—the state-of-the-art probabilistic imputation method.
>
>  As *BayOTIDE* is a factorization-based method, the model size is determined by the number of components. Corresponding to the hyper-parameter setting in Table 4, we set $D_r + D_s = 40$ components. Then the weights U's size is $40 \times 213$, and the components V's size is $40 \times 500 \times 2$(the mean and diag var). Thus, the total model size is $40 \times (213+500 + 500) = 48520$.
>
> *CSDI* is a diffusion-based method, the model size training memory is determined by the number of diffusion steps and the structure of the deep-based score estimator. The default structure *CSDI*'s(Figure 6 in CSDI paper) includes a spatial-temporal transformer with 5 layers and 8 heads and fully connected layes. The parameters also include the embedding(dim=128) of timestamps and features, and the diffusion step embedding. Thus, the total model size is at least $5 \times 16 \times 2 \times (500\times3+128\times3) + 128 \times (213 + 500) =  388224$.
>
> Thus, the model size of *BayOTIDE* is much smaller than *CSDI*. The performance gain of *BayOTIDE* over *CSDI* is mainly due to the factorization-based formulation, not the model size. In the *TIDER* paper[2], which takes a similar factorization formulation with *BayOTIDE*, the authors reported the memory usage of TIDER and CSDI (Figure 4 in TIDER paper). The memory usage of TIDER is much smaller than CSDI.
>
> [1]: Tashiro, Yusuke, et al. "Csdi: Conditional score-based diffusion models for probabilistic time series imputation." Advances in Neural Information Processing Systems 34 (2021): 24804-24816.
>
> [2]: LIU, SHUAI, et al. "Multivariate Time-series Imputation with Disentangled Temporal Representations." The Eleventh International Conference on Learning Representations. 2022.

---

> > ### Author Response · Authors · 2023-11-23
> >
> > Dear Reviewer SiJc,
> >
> > We greatly appreciate the time you took to review our paper. Due to the short duration of the author-reviewer discussion phase, we would appreciate your feedback on whether your main concerns have been adequately addressed. We are ready and willing to provide further explanations and clarifications if necessary.
> >
> > Thank you very much!

---

### Official Review · Reviewer_CYA1 · 2023-11-01

**Soundness:** 2 fair
**Presentation:** 1 poor
**Contribution:** 2 fair
**Rating:** 5
**Confidence:** 2

**Summary:**

This paper highlights the limitations of conventional time-series data imputation methods, which often disregard global trends, presume consistent sampling intervals, and are constrained to offline processing. To address these shortcomings, the authors introduce BayOTIDE, a groundbreaking imputation approach tailored for irregularly sampled data. Central to BayOTIDE's methodology is the interpretation of time series as amalgamations of low rank temporal factors, harnessing Gaussian Processes with varied kernels. By adeptly transitioning these processes into a state space model using stochastic differential equations, the method ensures computational efficiency and real time inference capabilities.

**Strengths:**

1. The paper tackles prevalent issues in time-series imputation, such as neglecting global trends, assumptions of regular sampling, and offline-only operation.

2. The introduction of treating time series as combinations of low-rank temporal factors is a novel perspective.

3. The transformation of Gaussian Processes into a state-space model using stochastic differential equations ensures the method is computationally efficient, which is crucial for real-world applications.

**Weaknesses:**

1. My main concerns regard writing/presentation and theoretical results.

2. The writing is rough, with some unclear and insufficient descriptions.

3. Unclear theoretical support.

**Questions:**

1. The paper requires improved structuring, particularly in terms of presenting the supporting theoretical guarantees.

---

> ### Author Response · Authors · 2023-11-21
>
> Thanks for your comments. Here are our responses. C: comments; R: response.
>
> > C1:"unclear and insufficient descriptions"
>
> R1: Could you please specify which part is unclear and insufficient? We will gladly add statements to clarify the unclear and insufficient parts.
>
> > C2:  "Unclear theoretical support."
>
> R2: The factorization-based formulation of the proposed method is aligned with Bayesian PCA[1], which is a well-known and classical Bayesian method with theoretical support.  The state-space GP prior[2] is known to be highly flexible/expressive for function estimation and has been used in numerous applications, including time series modeling [3]. The EP-based inference[4] is also a well-known approximate inference method with theoretical gaurentees on provable convergence. Thus, the proposed method is well-supported by the theoretical results of the above methods.
>
> Ref:
>
> [1] Bishop, Christopher. "Bayesian PCA." Advances in neural information processing systems 11 (1998).
>
>
> [2] Rasmussen, Carl Edward. "Gaussian processes in machine learning." Summer school on machine learning. Berlin, Heidelberg: Springer Berlin Heidelberg, 2003. 63-71.
>
>
> [3] Roberts, Stephen, et al. "Gaussian processes for time-series modeling." Philosophical Transactions of the Royal Society A: Mathematical, Physical and Engineering Sciences 371.1984 (2013): 20110550.
>
>
> [4]:Minka, Thomas P. "Expectation propagation for approximate Bayesian inference." arXiv preprint arXiv:1301.2294 (2013).

---

> > ### Author Response · Authors · 2023-11-21
> > **Response to Reviewer CYA1 before the end of discussion**
> >
> > Dear Reviewer CYA1,
> >
> > Since the End of author/reviewer discussions is just in one day, may we know if our response addresses your main concerns? If so, we kindly ask for your reconsideration of the score.
> >
> > Should you have any further advice on the paper and/or our rebuttal, please let us know and we will be more than happy to engage in more discussion and paper improvements. We would really appreciate it if our next round of communication could leave time for us to resolve any of your remaining or new questions.
> >
> > Thank you so much for devoting time to improving our work!

---

> > > ### Comment · Reviewer_CYA1 · 2023-11-22
> > >
> > > I appreciate the authors' detailed response and their efforts to address the points raised in my initial review. Their clarifications have provided a clearer understanding of the novelty and methodology of the work, which is indeed promising. Consequently, I have decided to change my rating accordingly to "marginally below the acceptance threshold".
> > >
> > > I believe that the paper, as it currently stands, still falls short of the standards required for acceptance. While the method shows potential, it requires further refinement in presentation and evaluation. Therefore, at this stage, I cannot recommend acceptance.

---

> > > > ### Author Response · Authors · 2023-11-23
> > > >
> > > > We thank you for reading our rebuttal and give feedback. We will definitely continue to improve our paper. As you mentioned you plan to raise your score to 5, it seems it is still 3. Could you please take a double check? Thank you again for the review work!

---

### Official Review · Reviewer_FPUc · 2023-11-04

**Soundness:** 4 excellent
**Presentation:** 3 good
**Contribution:** 3 good
**Rating:** 8
**Confidence:** 4

**Summary:**

This paper proposes a data imputation framework of multivariate nonstationary time series. The framework follows the classical Bayesian PCA (BPCA)-like imputation technique with the exception that the prior distribution is designed so the trend and seasonal components are captured.

Specifically, the model assumes the observation at each time point to be a linear combination of a few static basis vectors, where the coefficients of the linear combination are time-dependent. To allow seasonal and trend decomposition, the authors introduce specific prior distributions in the form of the Gaussian process (GP), where the temporal correlation is represented with the kernel function.

Although the inference procedure is analytically intractable, the authors leverage a variational Bayes approximation and derive a closed-form online updating equation.

**Strengths:**

- Solid formulation.
- Derivation of an analytic form of online updating equation for data imputation/model updates.
- The capability of splitting the trend and seasonal components, which is actually not straight forward when nonlinear temporal correlations are considered.


This is a good work. To the best of my knowledge, the framework is new. The basic concept of the BPCA-based imputation approach has been known for decades, but the paper adds a few new elements.

I vote for accepting the paper.

**Weaknesses:**

- Section 3.2 does not seem to play any role. Perhaps this paper has been rejected before, and the authors just wanted to add a "modern"-looking section. I got it. But it looks hardly related.
- Very poor proof-reading quality. Basic latex commands such as \eqref, \cite, etc. are not properly used. I know this might be re-re… submission, but PLEASE be respectful to the reviewers by meticulously proof-reading the manuscript before submission.

**Questions:**

- How does the well-known non-identifiability with respect to the unitary transformation in the UV-factorization form (1) take effect on the result?
- I am not clear how the almost linear trend could be separated in the result presented in Fig.1. How did the kernel expansion with the GPs produce the linear-looking trend component? What is the intuition behind it?
- Although I support accepting this paper, I am not 100% sure about the novelty. Bayesian PCA-based imputation is well-established. I know the main novelty comes from the time-series part, but your paper was not very clear about the "delta" from the pre-deep learning imputation works. To defend your work, please elaborate on the novelty in light of existing works. I just want to help you --- I suspect many ICLR reviewers do not have a strong understanding of the machine learning basics such as Bayesian PCA, and hence, papers like this one tend to receive unfairly low ratings.

---

> ### Author Response · Authors · 2023-11-21
>
> We thank the reviewer for the careful review and constructive suggestion—especially the suggestion on highlighting and elaborating the novelty of the proposed method in theoretical and methodological aspects.
>
> **We follow your suggestion and make a response overview to highlight the novelty and contribution of the proposed method, which is the top-level post in this forum. We regularly refer to the response overview**. We address the comments below:(C: comments; R: response.)
>
> >C1:" Section 3.2 does not seem to play any role. It looks hardly related."
>
> R1: The state-space GP we introduced in section 3.2  actually plays a key role in the proposed method. We use that state-space GP to model the trend and seasonal components in a continuous and functional manner with linear time cost. The chain structure of the state-space GP also enables online learning with KF. Specifically, we have to use the chain-structured prior (eq 2 in section 3.2) to replace the full GP prior (eq 7 in section 4.2) in the proposed method (the paragraph under eq7). We apologize for the unclear presentation and will add statements to highlight the role of the state-space GP in the proposed method.
>
> >C2: "Improve presentation quality"
>
> R2: Thanks for the suggestion! We will polish the presentation quality in the revised version. Actually, it's the first time we submit this paper to a peer-reviewed conference, so we are pretty grateful to get your valuable feedback and will continue to improve the presentation quality in the future :)
>
>
> >C3:  "How does the well-known non-identifiability with respect to the unitary transformation in the UV-factorization form (1) take effect on the result?"
>
> R3: Good point! We do acknowledge that the non-identifiability of unitary transformation will make learned components(V) and weights(U) lose the nice properties of uniqueness. However, we argue that the loss of non-identifiability is not a big issue in our case. The reason is that the components(V) is assigned with functional priors (GP), and are doomed to be non-linear and periodic. Then, even the non-identifiability effect takes place, saying the components(V) do not match the right "scale" but still have the right "shape", the weights(U) will be able to compensate for the "scale" effect. Actually, in our simulation study(Section 5.1), we do observe the non-identifiability effect, but when combining learned components(V) and weights(U), we could still get the right trend and seasonal components with the weights (Figure 1, (b)-(e)). We will add statements to clarify this point in the revised version.
>
> >C4: "I am not clear how the almost linear trend could be separated in the result presented in Fig.1. How did the kernel expansion with the GPs produce the linear-looking trend component? What is the intuition behind it?"
>
> R4: Good question! The short answer is that the separation of linear trend is not learned by the GP, but by the weights(U). Specifically, we set four GP components for $V(t)$ in the simulation study, and the first component is a Matern kernel, and the last three components are periodic kernels. Thus, the separated linear trends shown in Figure 1 (b)-(e) are actually from one GP kernel (Matern kernel), but with four different weights(U). The intuition behind it is that the different channels of dynamics may share similar trend components but with different weights. The message-passing algorithm in the proposed method is able to learn the channel-wise weight precisely with conditional moment-matching.
>
> >C5: "Elaborate Bayesian PCA-based imputation and highlight novelty compared with existing methods."
>
> R5: Great point! We just follow your suggestion and add general statements to highlight the novelty of the proposed method in the response overview. Please refer to sections **Novelty and contribution on time series imputation side** and  **Novelty and contribution on Bayesian learning side** in the response overview.

---

> ### Comment · Reviewer_FPUc · 2023-11-21
> **Thanks for clarifications.**
>
> "Overview of Author's Response and Clarifications" is well-structured and helped me a lot understand the overall picture of the novelty claimed. Although I'm still trying to internalize some aspects of the theory, I think the authors have provided enough clarifications for me to keep the original review and rating.

---

> > ### Author Response · Authors · 2023-11-21
> >
> > Thank you so much! We really appreciate your time to read our responses carefully and for your positive comments.

---

### Author Response · Authors · 2023-11-21
**Overview of Author's Response and Clarifications**

We thank all reviewers for their careful reading and detailed and considerate feedback. We are glad to see that almost all reviewers agree that **BayOTIDE** is a well-motivated work with solid formulation.

The following are the general clarifications for the reviewers' comments. We also provide detailed responses to each comment in the following sections.

### **1. Novelty and contribution on time series imputation side**

Though the factorization-based imputation methods are widely applied in general machine learning, like Bayesian PCA, tensor decomposition, and matrix factorization, they are rarely applied in time series imputation. To our knowledge, the most recent work to apply this for time series imputation is **TIDER**[Liu et al., ICLR 2022], which is a matrix factorization-based formulation. The gap there is because:

- Traditional time series models care more on capturing the temporal dependency, instead of the **cross-channel dependency** and **low-rank structure**.

- Directly applying factorization-based imputation to time series imputation is not feasible. The time series data is with **inherent temporal continuity**, which is not considered in the factorization-based methods. For example, directly applying Bayesian PCA to time series can not guarantee the learned factors are smooth over time mode.
Even the most recent work **TIDER**, which is a matrix factorization-based formulation, can not guarantee the temporal continuity of the learned trend factors, nor extend to a continuous and functional manner.

- In short, **it's non-trivial to model a low-rank factorization with constraints of temporal continuity** for applying classical imputation methods in multi-var time series.

Thus, the proposed method **BayOTIDE** is the **first work to perfectly bridge the gap of low-rank representation and temporal continuity under a probabilistic framework**. Firstly, the factorization imputation ensures its scalability and efficiency, even for time series with thousands of channels. Secondly, the usage of state-space GP naturally models the temporal dependency in a continuous and functional manner, even allowing Bayesian online inference.


### **2. Novelty and contribution on the Bayesian learning side**

We acknowledge that the fundamental
Bayesian learning tools used in **BayOTIDE**, like state-space GP, and conditional moment-matching, are mature techniques. However, they do not naturally fit in time series imputation. Specifically,

- The state-space GP, along with the Kalman filter, is designed for cases where **one observation corresponds one latent dynamic**. However, in BayOTIDE, **one observation corresponds to multiple latent dynamics(components)**.

- The CEP, is originally designed for cases where we observe **one observation** and then do **entry-wise conditional moment matching**. However, in BayOTIDE, as we observe **batch of observations** every time(values at different channels), we need to extend it to handle **batch-wise conditional moment matching**.

Thus, we have modified the original CEP to handle the batch-wise moment-matching (eq 9, 32), merging the approx. messages from different channels (eq 10, 11), and then feed latent message factors—aligned with the "observations" in standard state-space model/KF— to the component-wise Kalman filter (eq 12), and update the multiple latent dynamics in a synchronous way. To our knowledge, this is the **first online Bayesian inference framework for multivariate time series**, and it's **quite different from the original CEP and state-space GP, nor any existing Bayesian learning methods**


###  **3. Experiment Setting**

We thank all the reviewers for their suggestions on enhancing the experiment setting with more baselines, metrics, and
 ablation studies. We do agree that the experiment setting can be further improved.

 However, we would like to point out that the **main contribution of our work is on the methodological and formulation side**, which is the **first online Bayesian method** for multivariate time series. We follow the most recent work in this field: **CSDI** [ Tashiro, Neurips 2021],  **CSBI** [Chen, ICML 2022],  **TIDER** [Liu, ICRL 2023] to set up the experiment setting,  and think that the comprarion with the most **offline baselines** is sufficient to show the **effectiveness of the BayOTIDE**.

 We also add some new experiments based on the reviewers' comments:

 - The comparison with the single state-space model.
 - The comparison with linear + periodic kernel multi-output GP.
 - The numerical comparison of the model size between **BayOTIDE** and **CSDI**.

---

### Author Response · Authors · 2023-11-21
**Invitation for a Discussion**

We appreciate all reviewers' time and efforts in evaluating our work! In view of the limited available time, we would kindly like to ask the reviewers to please engage in a discussion with us (if not) given the submitted rebuttals so we can respond to the possible new questions in time.

---

### Meta-Review · Area_Chair_TFMP · 2023-12-05

**Metareview:**

This article presents an online Bayesian approach for imputation in time series. The approach is based on a latent decomposition into trends and seasonality factors. The paper offers a novel framework on a important practical problem. However, three of the reviewers emphasized the fact that the paper contained many typos, presentation issues and that comparisons to some baselines were missing. While the authors partially addressed these concerns in their responses and revised version, I agree with the majority of the reviewers that this article is not yet ready for publication.

**Justification For Why Not Higher Score:**

Need more polishing, and further review of the paper after the addition of the new experiments

**Justification For Why Not Lower Score:**

N/A

---

### Decision · Program_Chairs · 2024-01-16

Reject